# Modeling enhanced firn densification due to strain softening

Falk M. Oraschewski[1,2] and Aslak Grinsted[2]

[1]Department of Geosciences, University of Tübingen, Tübingen, Germany.
[2]Physics of Ice, Climate, and Earth, Niels Bohr Institute, University of Copenhagen, Copenhagen, Denmark.

**Correspondence:** Falk M. Oraschewski (falk.oraschewski@uni-tuebingen.de)

**Abstract.** In the accumulation zone of glaciers and ice sheets snow is transformed into glacial ice by firn densification. Classically, this process is assumed to solely depend on temperature and overburden pressure which is controlled by the accumulation rate. However, exceptionally thin firn layers have been observed in the high-strain shear margins of ice streams. Previously, it has been proposed that this firn thinning can be explained by an enhancement of firn densification due to the effect of strain softening inherent to power-law creep. This hypothesis has not been validated, and the greater firn densities in the presence of horizontal strain rates have not yet been reproduced by models. Here, we develop a model that corrects the firn densification rate predicted by classical, climate-forced models for the effect of strain softening. With the model it is confirmed that strain softening dominates the firn densification process when high strain rates are present. Firn densities along a cross section of the North-East Greenland ice stream (NEGIS) are reproduced with good agreement, validating the accuracy of the developed model. Finally, it is shown that strain softening has significant implications for ice core dating and that it considerably affects the firn properties over wide areas of the polar ice sheet, even at low strain rates. Therefore, we suggest that, besides temperature and accumulation rate, horizontal strain rates should generally be considered as a forcing parameter in firn densification modeling.

## 1 Introduction

Firn densification refers to the transformation of snow into glacial ice, which occurs in the uppermost layers of ice sheets and glaciers within their accumulation zones, when old snow, now denoted as firn, is buried under younger snow. The overburden pressure gradually increases and causes densification of the firn. Large scale ice flow is not considered by firn models even though it is known that ice is a non-newtonian material where strain reduces the viscosity (Goldsby and Kohlstedt, 2001). In this paper, we demonstrate that this effect can have a substantial impact on firn densification.

Firn densification is conventionally divided into stages where different physical mechanisms dominate. Initially, the Newtonian grain-boundary sliding is dominant for densities of $\rho \leq 550 \, \mathrm{kg \, m^{-3}}$, which is referred to as stage 1 of firn densification (Alley, 1987). In stage 2, at densities of $550 \, \mathrm{kg \, m^{-3}} \leq \rho \leq 830 \, \mathrm{kg \, m^{-3}}$, the non-Newtonian dislocation creep, also known as power-law creep, dominates the densification process until bubble close-off (BCO) (Maeno and Ebinuma, 1983). Beyond this point, defined as the firn-ice transition, the further densification is slowed as the enclosed gas-bubbles get compressed and eventually diffuse into the ice matrix (Salamatin et al., 1997).

For a variety of glaciological studies, properties of the firn need to be known. For example, it is essential to know the firn air content for deriving the mass balance of an ice sheet from changes of its surface elevation, measured by satellite altimetry (e.g. Helsen et al., 2008; Horlings et al., 2021). Ice core studies on the other hand require knowledge of the age difference $\Delta$age between the water and the gas records in the ice core. This age difference is primarily determined by the age of the firn at the lock-in depth, where the air in the firn pores loses contact with the atmosphere (e.g. Schwander et al., 1997; Buizert et al., 2015). The age at the lock-in depth is again closely related to the BCO age of the firn at the firn-ice transition.

In many applications, these properties are determined by employing a firn densification model. Over the years, a wide range of models have been developed (e.g. Herron and Langway, 1980; Alley, 1987; Arnaud et al., 2000; Arthern et al., 2010), which are adapted for different climatic conditions and application areas (Lundin et al., 2017). Stevens et al. (2020) provided with the Community Firn Model (CFM) an open-source, modular model framework, which comprises the most established firn densification models within a one-dimensional Lagrangian modeling scheme and allows the implementation of additional firn processes as modular extensions.

While efforts are being made to directly model the physical processes that lead to densification of the firn (Alley, 1987; Arnaud et al., 2000; Arthern et al., 2010), empirically tuned models, such as the Herron-Langway model (HL, Herron and Langway, 1980), are widely used. In this type of model, the densification rate equations are derived according to a few initial assumptions and subsequently tuned to fit density data from firn cores. Despite the different approaches, the majority of the existing models have in common that they merely consider temperature and accumulation rate as variable input parameters (Lundin et al., 2017). Only a few models take additional input parameters such as the impurity content into account (Bréant et al., 2017). Therefore, the classical models will be denoted as climate-forced in the following.

The limitation to climatic forcing is not only insufficient for finding firn model tuning parameters that are applicable for the whole Greenland ice sheet (Simonsen et al., 2013), but it also contradicts with observations of stress, or the corresponding strain, affecting the densification of firn (Zumberge et al., 1960; Crary and Wilson, 1961; Gow, 1968; Kirchner et al., 1979; Alley and Bentley, 1988; Riverman et al., 2019). As a consequence, existing firn models cannot sufficiently represent the reduced firn thickness that occurs in the high-strain shear margins of ice streams. However, the firn dynamics are important for understanding the dynamics of ice streams, as they potentially control their stability and contribute to the formation of shear margins troughs (Christianson et al., 2014; Riverman et al., 2019). The effect of horizontal strain rates on firn compaction moreover affects the firn air content in regions with strong ice dynamics and needs to be considered in mass balance studies (Horlings et al., 2021).

The reduced firn thickness due to strain is explained by two processes: horizontal divergence and strain softening.

Horizontal divergence of velocities causes a simple horizontal stretching and thus a vertical thinning of the firn. This effectively reduces the overburden load. The firn density itself is however not directly affected. Morris et al. (2017) accounted for this thinning in their firn model by introducing a correction factor that scales the vertical compaction strain rate according to the expected vertical thinning. A similar layer-thinning scheme was implemented into the CFM by Horlings et al. (2021) as an optional module.

Strain softening was first suggested by Alley and Bentley (1988) to accelerate the firn densification in the presence of horizontal strain rates. Alley and Bentley explain it by an enhancement of power-law creep, which as a non-Newtonian (pseudo-plastic) process scales with the square of the effective stress. Hence, a stress applied in horizontal direction can also accelerate the firn densification in vertical direction by reducing the effective viscosity. Strain softening mainly affects stage 2 of firn densification as this is where power-law creep is dominant (Alley and Bentley, 1988). This interpretation was supported by
Riverman et al. (2019), who observed that the variation of firn thickness, recorded along a cross-section of the North-East Greenland Ice Stream (NEGIS) by active seismic surveying, correlates with the net strain along the flow path of the past 400 years.

  Despite the observational evidence, firn densification models used for polar ice sheets do not capture the effect of strain softening. Based on a constitutive equation for the power-law creep of porous media (Duva and Crow, 1994), Gagliardini
and Meyssonnier (1997) have developed a glacier flow model that inherently considers compaction – and in particular strain softening – of the firn. It is widely used in studies of alpine glaciers (Lüthi and Funk, 2000; Zwinger et al., 2007; Licciulli et al., 2020), but has not been applied to the polar ice sheets. The reasons are presumably that this approach is computationally more expensive on large scales, more difficult to implement and that the range of conditions, that this model is calibrated and tested on, is not as wide as it is for the classical firn models.

In this paper, we aim to model the effect of strain softening in the firn with a different approach. We derive a scale factor that can be applied with any climate-forced firn densification model to correct its predicted firn densification rate for the impact of strain. Our approach is computationally cheap, simple to implement and thereby can extend the application range of the well-established classical firn models even further.

## 2 Theory

A firn densification model in a Lagrangian formulation expresses the densification rate of a firn layer as a function of external forcing parameters and internal parameters, representing its current state. The external parameters are generally time-variable. In a climate-forced model these are the temperature $T$ and the accumulation rate $\dot{b}$ or the overburden load $\sigma$, derived from $\dot{b}$. As an internal parameter most firn models only consider the current density $\rho$ of the firn layer. Additionally, for newly formed snow layers at the surface the initial snow density $\rho_0$ is required as a boundary condition, which in most applications is assumed
to be a site-specific constant. The densification rate of a climate-forced model, here denoted with the subscript c, is hence given in the form of

$$\left(\frac{\mathrm{D}\rho}{\mathrm{D}t}\right)_{\mathrm{c}} = f(T, \sigma, \rho), \tag{1}$$

with the time $t$. As such climate-forced models however neglect the effects of horizontal divergence and strain softening, the given densification rate will differ from its actual value.

Generally, at a specific point in the firn the state of the strain rates is described by the symmetric strain rate tensor

$$\dot{\boldsymbol{\varepsilon}} = \begin{pmatrix} \dot{\varepsilon}_{xx} & \dot{\varepsilon}_{xy} & \dot{\varepsilon}_{xz} \\ \dot{\varepsilon}_{xy} & \dot{\varepsilon}_{yy} & \dot{\varepsilon}_{yz} \\ \dot{\varepsilon}_{xz} & \dot{\varepsilon}_{yz} & \dot{\varepsilon}_{zz} \end{pmatrix}, \tag{2}$$

which consists of two normal horizontal strain rate components ($\dot{\varepsilon}_{xx}$, $\dot{\varepsilon}_{yy}$), the normal vertical strain rate component ($\dot{\varepsilon}_{zz}$) and three shear components ($\dot{\varepsilon}_{xy}$, $\dot{\varepsilon}_{xz}$, $\dot{\varepsilon}_{yz}$). Following Morris et al. (2017), its trace defines the volumetric strain rate $\dot{\varepsilon}_{\mathrm{vol}} = \mathrm{tr}(\dot{\boldsymbol{\varepsilon}})$ which in firn is related to the densification rate according to

$$\dot{\varepsilon}_{\mathrm{vol}} = -\frac{1}{\rho}\frac{\mathrm{D}\rho}{\mathrm{D}t}. \tag{3}$$

In the case of climate-forced models all components of $\dot{\boldsymbol{\varepsilon}}$ except for the vertical strain rate are assumed to be zero and the strain rate tensor reads as

$$\dot{\boldsymbol{\varepsilon}}_{\mathrm{c}} = \begin{pmatrix} 0 & 0 & 0 \\ 0 & 0 & 0 \\ 0 & 0 & \dot{\varepsilon}_{zz,\mathrm{c}} \end{pmatrix}. \tag{4}$$

Thereby, not only the volumetric strain rate is reduced to the vertical strain rate $\dot{\varepsilon}_{zz,\mathrm{c}}$, but also the latter will differ from the
general case, meaning that $\dot{\varepsilon}_{zz,\mathrm{c}} \neq \dot{\varepsilon}_{zz}$. Eq. 3 for the climate-forced model case then takes the form of

$$\dot{\varepsilon}_{zz,\mathrm{c}} = -\frac{1}{\rho}\left(\frac{\mathrm{D}\rho}{\mathrm{D}t}\right)_{\mathrm{c}}. \tag{5}$$

Here, we aim to derive the actual densification rate $\mathrm{D}\rho/\mathrm{D}t$ as a function of the densification rate given by a climate-forced model $(\mathrm{D}\rho/\mathrm{D}t)_{\mathrm{c}}$. The derivation is complicated by the fact that when horizontal divergence is active alongside strain softening, both effects are entangled. This is the case when the normal horizontal strain rates do not balance, i.e. $\dot{\varepsilon}_{\mathrm{h}} = \dot{\varepsilon}_{xx} + \dot{\varepsilon}_{yy} \neq 0$.
To simplify the situation, we assume that horizontal divergence only has a negligible influence on the pressure, which is justified for the second firn stage, where already a significant overburden pressure has built up and the pressure component by horizontal compression can be neglected. With this assumption both effects can be taken as being independent of each other and we can correct for them separately. This we do by first applying the strain softening correction that is derived in the following and then correcting for horizontal divergence subsequently, using the layer thinning scheme by Morris et al. (2017)
and Horlings et al. (2021).

For the derivation of the strain softening correction, the above assumption means that $\dot{\varepsilon}_{\mathrm{h}} \ll \dot{\varepsilon}_{zz}$ and hence the volumetric strain rate in Eq. 3 is approximated by the vertical strain rate, so that $\dot{\varepsilon}_{\mathrm{vol}} \approx \dot{\varepsilon}_{zz}$. We obtain

$$\frac{\mathrm{D}\rho}{\mathrm{D}t} = -\dot{\varepsilon}_{zz}\rho, \tag{6}$$

which describes how a strain softening corrected vertical strain rate $\dot{\varepsilon}_{zz}$ translates back into a corresponding corrected densifi-
cation rate.

By combining Eqs. 5 and 6 it then follows that

$$\frac{\mathrm{D}\rho}{\mathrm{D}t} = \frac{\dot{\varepsilon}_{zz}}{\dot{\varepsilon}_{zz,\mathrm{c}}} \left(\frac{\mathrm{D}\rho}{\mathrm{D}t}\right)_{\mathrm{c}}. \tag{7}$$

Hence, the densification rate output of a climate-forced firn model can be corrected for the effect of strain softening by multiplication with a scale factor that is given by the ratio of the corresponding vertical strain rate components. We aim in the following to determine this scale factor.

## 2.1 Constitutive relations for firn

Firn is a compressible material. Following Duva and Crow (1994), its deformation is given by two coupled constitutive relations which represent a rigid reinforcing phase and a compressible phase, that are not to be confused with the ice and air phases often used for describing firn. The rigid phase is defined by the tensorial relation between deviatoric stress and deviatoric strain rate, as given by Glen's flow law for the deformation of incompressible ice (Glen, 1955; Nye, 1957). The scalar relation of the compressible phase is set between the isotropic pressure and the volumetric strain rate.

While the proportionality factors between stress and strain, also denoted as effective viscosities, in the two constitutive relations generally differ, they are coupled in non-Newtonian materials by the fact that the strain rates of one phase also reduce the effective viscosity of the other. Duva and Crow (1994) formulated a constitutive relation for compressible power-law creeping materials and took account of the coupling by combining the relations of the two phases with corresponding density dependent weighting coefficients $a$ and $b$. The ratio between these two factors essentially determines by how much the additional deviatoric strain rates reduce the effective viscosity of the compressible phase.

For firn the two weighting coefficients have been calibrated by Gagliardini and Meyssonnier (1997) using firn density data of Site 2, Greenland. Unfortunately, the calibration is ill-posed and exactly the ratio of the weighting coefficients was fixed to solve the issue (Gagliardini, 2012). The coefficients $a$ and $b$ are therefore poorly constrained and in turn a scale factor based on the model by Gagliardini and Meyssonnier (1997) would also be. We take in the following a different approach by assuming that the effective viscosities of the two phases are equally affected by strain softening. This is feasible as we are not interested in the total effective viscosity of the compressible phase, but only in its relative change.

## 2.2 Compaction on the microscale

To motivate this approach and derive the scale factor, we look at firn compaction on a microscopic level, where firn is a mixture of solid ice and air. Internal stresses are concentrated at the contact areas between grains and stress chains form which carry the overburden load (Peters et al., 2005). In the second firn stage, where grains have settled, the rate of compaction is controlled by how fast these chains deform.

The deformation of the solid ice matrix can be described by Glen's flow law (See also Fourteau et al., 2020),

$$\dot{e}_{\mathrm{ice},ij} = A\tau_{\mathrm{ice,E}}^{n-1}\tau_{\mathrm{ice},ij}, \tag{8}$$

with the deviatoric strain rate $\dot{e}_{\text{ice}}$, the deviatoric stress $\tau_{\text{ice}}$ and the effective deviatoric stress $\tau_{\text{ice,E}}$, which is the second invariant of the deviatoric stress tensor. $A$ is the creep factor, which represents the temperature impact and $n$ the creep exponent. As power-law creep is dominant in the second firn stage, we assume that other deformation processes can be disregarded.

Glen's flow law can equivalently be formulated in terms of the strain rate dependent effective ice viscosity $\eta_{\text{ice}}$

$$\dot{e}_{\text{ice},ij} = \frac{1}{2\eta_{\text{ice}}}\tau_{\text{ice},ij}$$

$$\eta_{\text{ice}} = \left[2A^{1/n}\dot{\varepsilon}_{\text{ice,E}}^{m}\right]^{-1}, \tag{9}$$

with the exponent $m = 1 - 1/n$. The effective ice viscosity depends on the effective strain rate $\dot{\varepsilon}_{\text{ice,E}}$, which is a measure for the strength of deformation in the stress chains, given by the second invariant of the local deviatoric strain rate tensor.

In practice, it is difficult to apply Eq. 9 to densification as it would require detailed knowledge of the 3-dimensional structure of the ice-air matrix in order to relate how the overburden pressure is related to the deviatoric stress experienced throughout.

The overburden load tends to focus into stress chains such that the overburden is predominantly supported by relatively few ice grains (Peters et al., 2005). Most other grains will be shielded by these chains and will experience little deformation until it becomes their turn to carry the load.

To relate the microscopic deformation with the macroscopic firn compaction despite the difficulties, we need to make two simplyfing assumptions: We expect that high macroscale deformation and compaction correlates with strong microscale defor-

mation. Therefore, we assume that the microscale effective strain rate in the stress chains is proportional to the macroscopic effective strain rate $\dot{\varepsilon}_{\text{E}}$ of the strain rate tensor that contains compaction (Eqs. 2 or 4), such that

$$\dot{\varepsilon}_{\text{ice,E}} = k_1 \cdot \dot{\varepsilon}_{\text{E}}, \tag{10}$$

where $k_1$ is a scaling constant that we assume only depends on the material properties of the firn.

The second assumption is that the densification rate is controlled by the strain rate of the load-carrying grains as their

deformation must be the limiting factor for the densification. Hence, we assume that the bulk behavior of the firn scales with the behavior of such characteristic grains, as follows:

$$\dot{\varepsilon}_{ij} = k_2 \cdot \dot{e}_{\text{ice},ij}, \tag{11}$$

where $\dot{e}_{\text{ice},ij}$ represents a characteristic value of such grains, and $k_2$ is again a material dependent scaling constant.

Applying the second assumption to Eq. 9 gives us

$$\dot{\varepsilon}_{zz} = \frac{k_2}{2\eta_{\text{ice}}}\tau_{\text{ice},zz}, \tag{12}$$

where $\tau_{\text{ice},zz}$ is a characteristic deviatoric stress experienced by the grains that limit the deformation. We will assume that $\tau_{\text{ice},zz}$ only depends on load and the material properties of the firn, but is not affected by deformation in other directions.

Equations 8 to 12 are applicable independent of whether the additional strain rate components are considered. They can be formulated both in terms of the strain softening corrected model and the climate-forced model, with the only difference being

whether the macroscopic effective strain rate is computed from Eq. 2 or 4. In particular, $k_1$, $k_2$ and $\tau_{\text{ice},zz}$ do not differ between the two cases as they are assumed to be independent of the additional strain rate components. Thus, Eq. 12 can analogously be formulated for the climate-forced model case as

$$\dot{\varepsilon}_{zz,\text{c}} = \frac{k_2}{2\eta_{\text{ice,c}}}\tau_{\text{ice},zz}, \tag{13}$$

where the effective ice viscosity in the climate-forced model $\eta_{\text{ice,c}}$ will be higher than the actual effective ice viscosity $\eta_{\text{ice}}$.

## 2.3 The scale factor

By dividing Eqs. 12 and 13, the strain rate ratio in Eq. 7, which is the scale factor, can be expressed in terms of a solid ice viscosity ratio:

$$\frac{\dot{\varepsilon}_{zz}}{\dot{\varepsilon}_{zz,\text{c}}} = \frac{\eta_{\text{ice,c}}}{\eta_{\text{ice}}}. \tag{14}$$

Inserting the expression for the effective ice viscosity (Eq. 9) and using Eq. 10 gives

$$\frac{\dot{\varepsilon}_{zz}}{\dot{\varepsilon}_{zz,\text{c}}} = \left(\frac{\dot{\varepsilon}_{\text{ice,E}}}{\dot{\varepsilon}_{\text{ice,E,c}}}\right)^m = \left(\frac{\dot{\varepsilon}_{\text{E}}}{\dot{\varepsilon}_{\text{E,c}}}\right)^m. \tag{15}$$

Thus, the enhancement from strain softening can be expressed in terms of macroscopic firn strain rates. By expanding the effective strain rates into components, we get

$$\frac{\dot{\varepsilon}_{zz}}{\dot{\varepsilon}_{zz,\text{c}}} = \left(\frac{\dot{\varepsilon}_{xx}^2 + \dot{\varepsilon}_{yy}^2 + \dot{\varepsilon}_{zz}^2 + 2\dot{\varepsilon}_{xy}^2 + 2\dot{\varepsilon}_{xz}^2 + 2\dot{\varepsilon}_{yz}^2}{\dot{\varepsilon}_{zz,\text{c}}^2}\right)^{m/2}, \tag{16}$$

which can be rewritten in the form of

$$r_{\text{v}} = \left(r_{\text{h}}^2 + r_{\text{v}}^2\right)^{m/2}, \tag{17}$$

whereby the following two variables are defined:

$$r_{\text{v}} := \frac{\dot{\varepsilon}_{zz}}{\dot{\varepsilon}_{zz,\text{c}}}, \tag{18}$$

$$r_{\text{h}} := \left(\frac{\dot{\varepsilon}_{xx}^2 + \dot{\varepsilon}_{yy}^2 + 2\dot{\varepsilon}_{xy}^2 + 2\dot{\varepsilon}_{xz}^2 + 2\dot{\varepsilon}_{yz}^2}{\dot{\varepsilon}_{zz,\text{c}}^2}\right)^{1/2}. \tag{19}$$

Existing purely climate driven densification models, such as the classical HL model, provide us with an estimate of the vertical strain rate $\dot{\varepsilon}_{zz,c}$ under the assumption that all other components of the strain rate tensor are zero (Eq. 5). The remaining components of $r_{\text{h}}$ can be estimated from surface velocity observations or flow modeling. Their strength can be measured by the effective external strain rate $\dot{\varepsilon}_{\text{E,e}} = \left[\frac{1}{2}\left(\dot{\varepsilon}_{xx}^2 + \dot{\varepsilon}_{yy}^2\right) + \dot{\varepsilon}_{xy}^2 + \dot{\varepsilon}_{xz}^2 + \dot{\varepsilon}_{yz}^2\right]^{1/2}$.

In summary, the variable $r_{\text{v}}$ corresponds exactly to the scale factor that is sought. If all strain rate components in $r_{\text{h}}$ are known, only Eq. 17 is left to be solved for $r_{\text{v}}$ to obtain the scale factor for correcting the densification rate of a climate-forced

model for the total effect of strain softening. But even if only some of the external strain rate components in Eq. 19, e.g. the horizontal strain rates, are known, this approach can be used to correct for their contribution to strain softening enhancement of the densification rate.

The solution of Eq. 17 depends on the creep exponent. Dislocation creep is the key process driving densification in the second firn stage (Maeno and Ebinuma, 1983). In this paper we therefore use a creep exponent of $n = 4$, characteristic for

dislocation creep (Goldsby and Kohlstedt, 2001; Bons et al., 2018). In this case, Eq. 17 is essentially a solvable polynomial of order eight. Its solution is given by

$$
\kappa_1 = \left[ 9r_{\mathrm{h}}^8 + \sqrt{81r_{\mathrm{h}}^{16} + 768r_{\mathrm{h}}^{18}} \right]^{1/3},
$$

$$
\kappa_2 = \left[ 1 + 8r_{\mathrm{h}}^2 + \sqrt[3]{\frac{32}{9}}\kappa_1 - \frac{\sqrt[3]{\frac{8192}{3}}r_{\mathrm{h}}^6}{\kappa_1} \right]^{1/2},
$$

$$
\kappa_3 = \left[ \frac{1}{2} + 4r_{\mathrm{h}}^2 - \frac{\kappa_1}{\sqrt[3]{18}} + \frac{\sqrt[3]{\frac{128}{3}}r_{\mathrm{h}}^6}{\kappa_1} + \frac{1 + 12r_{\mathrm{h}}^2 + 24r_{\mathrm{h}}^4}{2\kappa_2} \right]^{1/2},
$$

$$
r_{\mathrm{v}} = \left[ \frac{1}{4} + \frac{1}{4}\kappa_2 + \frac{1}{2}\kappa_3 \right]^{1/2}. \tag{20}
$$

An alternative solution for the often used case of $n = 3$ is given in Oraschewski (2020).

The derivation of the strain softening enhancement above relies on a number of assumptions. We have assumed that in the

210 second stage firn densifies by dislocation creep and that this densification is driven by vertical compression. Other densification processes and horizontal compression are therefore assumed to be negligible and strain softening and horizontal divergence are assumed to be independent of each other. We have further assumed that the microscale solid ice deformation is the key process limiting the rate of firn densification. To relate this to the macroscale compaction we assume that the microscale effective strain rate of deformation scales with the macroscale effective strain rate that contains compaction (Eq. 10) and that firn strain rates

scale with a characteristic value for the solid ice strain rate (Eq. 11). The scaling constants ($k_1$ and $k_2$) and the characteristic vertical deviatoric stress of the rate-limiting grains ($\tau_{\mathrm{ice},zz}$) are assumed to have no directional dependence but can depend on density, microstructure, temperature, and load. From these assumptions follows that the effective viscosities in the constitutive equations of the rigid and the compressible phase in the firn are assumed to be equally affected when additional strain rates soften the firn. In this way, we could obtain a model for the enhancement of firn densification by strain softening that involves

zero free parameters.

## 2.4 Regularization

The densification in climate-forced models, such as the HL model, is characterized by an exponential decay towards the density of solid ice. The vertical strain rate $\dot{\varepsilon}_{zz,\mathrm{c}}$ in such models goes to zero as the firn density approaches the density of ice of $917\,\mathrm{kg\,m^{-3}}$. Thereby, $r_{\mathrm{h}}$ and with it the scale factor $r_{\mathrm{v}}$ go towards infinity. This singularity causes a nonphysical behavior

of the strain softening model, where the density of ice is approached almost instantaneously at a certain point.

To circumvent this issue, a regularization is introduced. Inspired by *regularized Glen's flow law* (Greve and Blatter, 2009, Ch. 4) a residual strain rate $\dot{\varepsilon}_0$ is added to the vertical strain rate to ensure a finite correction factor:

$$r_{\mathrm{h}} := \left( \frac{\dot{\varepsilon}_{xx}^2 + \dot{\varepsilon}_{yy}^2 + \dot{\varepsilon}_{zz}^2 + 2\dot{\varepsilon}_{xy}^2 + 2\dot{\varepsilon}_{xz}^2 + 2\dot{\varepsilon}_{yz}^2}{\left(\dot{\varepsilon}_{zz,\mathrm{c}} + \dot{\varepsilon}_0\right)^2} \right)^{1/2}. \tag{21}$$

Leaving the perspective of firn densification modeling, the residual strain rate can be associated with the general thinning of firn and ice layers in ice sheets that is induced by the flow of ice towards the ice sheet margins. While the densification part of the vertical strain rate approaches zero, this contribution remains finite. The residual strain rate can be obtained by flow modeling or measured using strain gauge instruments (Zumberge et al., 2002; Elsberg et al., 2004) or phase-sensitive radar (Gillet-Chaulet et al., 2011; Zeising and Humbert, 2021).

## 2.5 Tuning bias correction

Empirical climate-forced firn models contain two tuning parameters which represent the dependency and sensitivity of the densification rate to temperature and accumulation rate. They are obtained by tuning the model to firn density measurements from, for example, firn cores, whereby it is assumed that all inter-site density variability can be attributed to the variability of temperature and accumulation rate between the sites. If another process that is driven by a different forcing parameter also affects the densification, its contribution will be implicitly captured by the two tuning parameters for the climatic forcing. Thereby, the additional process is not only inadequately represented, but also the model sensitivity to the climatic forcing will be inaccurate. We refer to the implicit contribution as a tuning bias.

In Sect. 5 we show evidence that neglecting strain softening can cause such a tuning bias. If the firn cores that were used to tune the model were subjected to a mean effective external strain rate of $\overline{\dot{\varepsilon}_{\mathrm{E,e}}}$, it can be expected that the model outputs approximately capture a strain softening enhancement that corresponds to a strain rate forcing of that size. As a consequence, this contribution would be considered twice when our strain softening enhancement is applied to a tuning biased empirical firn model.

The densification rate output of the climate-forced model therefore actually has to be seen as the densification rate by climatic forcing alone $(\mathrm{D}\rho/\mathrm{D}t)_0$ times the scaling factor $r_{\mathrm{cor}}$ for strain softening due to an effective strain rate of $\dot{\varepsilon}_{\mathrm{cor}} = \overline{\dot{\varepsilon}_{\mathrm{E,e}}}$:

$$\left( \frac{\mathrm{D}\rho}{\mathrm{D}t} \right)_{\mathrm{c}} = r_{\mathrm{cor}} \left( \frac{\mathrm{D}\rho}{\mathrm{D}t} \right)_0. \tag{22}$$

As given in Eq. 7 our scale factor is applied to the biased densification rate of the climate forced model $(\mathrm{D}\rho/\mathrm{D}t)_{\mathrm{c}}$, whereas it is assumed that the input densification rate is unbiased. To correct for the tuning bias, the strain softening scale factor has to be divided by $r_{\mathrm{cor}}$, whereby Eq. 7 is replaced by

$$\frac{\mathrm{D}\rho}{\mathrm{D}t} = \frac{r_{\mathrm{v}}}{r_{\mathrm{cor}}} \left( \frac{\mathrm{D}\rho}{\mathrm{D}t} \right)_{\mathrm{c}}. \tag{23}$$

With this equation the strain softening model has no effect when the effective strain rate input $\dot{\varepsilon}_{\mathrm{E,e}}$ matches $\dot{\varepsilon}_{\mathrm{cor}}$. In the case of $\dot{\varepsilon}_{\mathrm{E,e}} < \dot{\varepsilon}_{\mathrm{cor}}$ the densification rate will be reduced.

Correctly determining $r_{\mathrm{cor}}$ is actually prevented by the fact that the unbiased densification rate $(\mathrm{D}\rho/\mathrm{D}t)_0$ is unknown. As a first-order approximation we therefore compute it based on the output of the climate forced model, analogously to the computation of the scale factor $r_{\mathrm{v}}$. Deriving the tuning bias correction from the biased model output itself will induce a small error into the also relatively small correction. In the overall picture, it will be a tiny error that can be neglected.

## 3 Application of the model

Our new model for strain softening (Eq. 20) takes the form of a scale factor to classical, climate-forced models. It allows us to calculate how much faster firn densifies when the firn pack is exposed to e.g. horizontal strain rates. The quality of the fit to the overall density profile at a single site is therefore a product of both the quality of the chosen classical density model and the quality of the applied scale factor. The scale factor is widely applicable to many different firn densification models. In this paper, we want to isolate and focus on the impact of the scale factor rather than the combined effect. Equation 7 models the change in densification rate. Thus, the validation of the model must also focus on whether it is able to reproduce density changes between different strain environments. We will therefore test the predictions of the strain enhancement model on data collected at NEGIS, where large variations in horizontal strain rates, and thus in $r_{\mathrm{h}}$, occur within a relatively small area.

### 3.1 Assumption of depth uniformity

For the application of the model, we assume in this paper that the horizontal velocities in the firn are uniform with depth. While, from a theoretical point of view, this assumption is not needed in the strain softening model, it is required due to the lack of internal velocity data. This has two consequences: first, horizontal strain rates ($\dot{\varepsilon}_{xx}$, $\dot{\varepsilon}_{yy}$, and $\dot{\varepsilon}_{xy}$) are also assumed to be independent of depth in the firn column. And second, the vertical shear rates ($\dot{\varepsilon}_{xz}$ and $\dot{\varepsilon}_{yz}$) are assumed to be negligible in comparison to the horizontal strain rates. These assumptions allow to estimate the numerator of $r_{\mathrm{h}}$ (Eq. 19) from surface velocities alone.

When the strain softening enhancement is applied to the interior of the Greenland (GrIS) and Antarctic ice sheets (AIS), as we do here, this is justified. In ice sheets vertical shear is predominantly confined to the deepest layers (e.g. Gundestrup et al., 1993; Weikusat et al., 2017), whereas firn only accounts for a small, upper fraction of the total ice thickness over the majority of the ice sheet. The change of horizontal velocities in the firn will therefore be negligible. This is also a common first approximation in ice core dating, as for example the Dansgaard and Johnsen (1969) model assumes that horizontal velocities are near uniform with depth in the upper half of the ice sheet.

Although the uppermost firn layers are softer and therefore more likely to entail vertical shear (See Schwerzmann, 2006, Fig. 5.2), this can also be neglected in the interior of the ice sheets. Due to their flat surface, a driving force is missing that can induce such shear. But even if vertical shear would occur in the top layers, this could safely be ignored in our model, because the very soft layers are located in firn stage 1, whereas strain softening first becomes important in firn stage 2.

As the main application area of the model are the polar ice sheets and not alpine glaciers, where the Gagliardini and Meyssonnier (1997) model is more suitable, the model is currently implemented such that only depth uniform horizontal strain rates are

considered, but nonetheless, it is principally possible to take nonuniform and vertical shear rates into account when they are known for the study site by e.g. bore hole deformation.

According to the assumption that the horizontal strain rates solely determine the strength of the strain softening, their magnitude will in the following be expressed by the effective horizontal strain rate $\dot{\varepsilon}_{\mathrm{E,h}} = \left[ \frac{1}{2} \left( \dot{\varepsilon}_{xx}^2 + \dot{\varepsilon}_{yy}^2 \right) + \dot{\varepsilon}_{xy}^2 \right]^{1/2}$, which replaces the effective external strain rate that was used before.

## 3.2 Implementation

The strain softening scale model is implemented as an optional module into the CFM by Stevens et al. (2020). Supported by its Lagrangian modeling scheme, the implementation itself is rather simple and computationally cheap. The uncorrected vertical strain rate is computed from the initial densification rate according to Eq. 5 using a classical densification model. In combination with the strain rate input data, the variable $r_{\mathrm{h}}$ can then be computed by Eq. 21, from which the correction factor of the vertical strain rate is computed with Eq. 20 and applied according to Eq. 7 or 23, depending on whether the tuning bias correction is applied.

In order to reduce the number of input parameters for the horizontal strain rate from three ($\dot{\varepsilon}_{xx}$, $\dot{\varepsilon}_{xy}$, $\dot{\varepsilon}_{yy}$) to two, they are loaded in the form of the principal horizontal strain rates (see e.g. Nye, 1959). In this way, the shear components disappear, such that the amount of input data is reduced without the loss of any relevant information.

The strain softening model is only applied in the second stage of firn densification for $\rho > 550 \, \mathrm{kg \, m^{-3}}$ as power-law creep is thought to dominate firn densification only in this range, while the Newtonian grain-boundary sliding is driving the densification process before. Although a smooth transition between the two processes over a range of densities is expected (Hörhold et al., 2011), its exact form is unknown and a sharp transition is assumed in our present implementation. Nonetheless, attempts to implement a smooth transition did not affect the model output significantly (not shown).

Within the CFM framework the strain softening model can be executed in combination with any of the implemented climate-forced firn densification models. In the following model experiments, we will use the Herron-Langway model (Herron and Langway, 1980) in its stress-based formulation as the input model. While the HL model on one hand is capable of reproducing the firn densities in the investigated NEGIS region accurately, its stress-based formulation additionally considers potential strain-induced changes of the overburden load, which an accumulation-based formulation would not capture.

Temperature evolution is neglected in our model experiments, as we aim to assess the general impact of strain softening on firn densification and thereby study processes occurring in the second firn stage, where temperature is approximately stable. At this depth seasonal temperature variations are dampened by heat conduction and only a recent warming trend remains, which for North Greenland lies on the order of $1^{\circ}\mathrm{C}$ (Orsi et al., 2017) and, hence, has a minor impact on firn densification.

## 4 Data

Modeling strain softening requires knowledge about the horizontal strain rates. For the modeling experiments conducted in this paper, horizontal strain rates are computed from surface velocity maps of the GrIS and AIS using the logarithmic strain

rate computation method as discussed by Alley et al. (2018). Nominal computation of the strain rates proved however to be sufficient as differences between the two computation methods were smaller than the uncertainty induced by the velocity data itself.

For the GrIS horizontal strain rates are computed from the MEaSUREs Multi-year Greenland Ice Sheet Velocity Mosaic (Joughin et al., 2016, 2018), which has a spatial resolution of $250\,\mathrm{m} \times 250\,\mathrm{m}$. The ice sheet margins are set following the MEaSUREs Greenland Ice Mapping Project (GIMP) Land Ice and Ocean Classification Mask (Howat, 2017; Howat et al., 2014). For the AIS the MEaSUREs InSAR-Based Antarctica Ice Velocity Map, Version 2 (Rignot et al., 2017, 2011; Mouginot et al., 2012) with a spatial resolution of $450\,\mathrm{m} \times 450\,\mathrm{m}$ is used.

Before determining the strain rates from the velocity fields, a Gaussian filter is applied on the velocity maps to reduce the impact of processing artifacts in the data, which likely were caused by combining velocity data from different sources for producing these velocity fields. These artifacts appear as a grid structure in the strain rate products and clearly do not represent any physical information, but would lead to an overestimation of the horizontal strain rates, if not removed. For the GrIS velocity data, a Gaussian filter with a standard deviation of $2\,\mathrm{pixels}$ is applied. For the AIS we use a variable Gaussian filter with standard deviations between $2-10\,\mathrm{pixels}$. Regions with poor data coverage and higher reported uncertainties — e.g. in the polar hole — are smoothed more. The smoothing reduces the effective spatial resolution of the velocity grid, which can dampen local strain maxima for example in ice stream shear margins. However, this is not a major concern, because during modeling km-scale variations of horizontal strain rates in any case average out over the firn age.

Using the strain rate products, the mean of the effective horizontal strain rates at the locations of the firn cores that were used to tune the HL model is determined, whereby Little America V site is excluded, as no data exist for this point. We obtain a value of $\dot{\varepsilon}_{\mathrm{cor}} = 4.5 \times 10^{-4}\,\mathrm{yr}^{-1}$, which we use for the tuning bias correction in the following.

For validating the strain softening model, firn density data recorded at the NEGIS in the vicinity of the EGRIP ice core site are used. As suggested by Vallelonga et al. (2014) the NEGIS with its high-strain shear margins offers excellent opportunities for studying firn densification processes. The data reproduced in this study comprise a $37\,\mathrm{km}$-long cross section of the NEGIS firn densities recorded with active seismic surveying by Riverman et al. (2019) and the density of the NEGIS firn core (Vallelonga et al., 2014). Their locations are shown in Fig. 1. Additionally, the density of the EGRIP S5 2019 shear margin firn core is modeled with the intent to compare the model with directly measured firn density data from a high-strain area. However, this data are unfortunately not available yet, as the firn core is stored at the EGRIP station and is not accessible, because of COVID-19-related restrictions of field work.

The horizontal strain rates that the firn at these sites has experienced in the past are computed by step-wise backtracing their position according to the velocity field with a monthly resolution and interpolating the computed horizontal strain rate components to these points at every time step.

We force the model with a constant temperature of $-29.9°\mathrm{C}$. This is the seasonality-corrected mean of the $10\,\mathrm{m}$ temperature recorded between June 2019 and January 2021 at the PROMICE weather station at EGRIP (Fausto and van As, 2019; Fausto et al., 2021). Using a constant temperature input is justified because we are mainly interested in firn processes occurring below a depth of $10\,\mathrm{m}$, where seasonal variability of temperature is smoothed out by heat conduction and the impact of the

355 general warming trend in Greenland is minor. Further, we do not expect a significant spatial variability of temperature over this relatively small study region.

As accumulation rate input we use the values derived by Riverman et al. (2019) from the Accumulation Radar of an Operation IceBridge flightline (Paden et al., 2014, updated 2018) crossing the NEGIS density profile, see Fig. 1b. We extrapolate the accumulation rate from the flight line to the sites where densification is modeled using nearest point interpolation. At NEGIS

the surface density is set to $295\,\mathrm{kg\,m^{-3}}$, which is the density measured in the top $10\,\mathrm{cm}$ in this region by Schaller et al. (2016). For the residual strain rate a value of $\dot{\varepsilon}_0 = -0.7 \times 10^{-4}\,\mathrm{yr^{-1}}$ is applied, following observation at EGRIP by Zeising and Humbert (2021).

The effect of strain softening on firn densification is not only studied on local, but also on ice sheet wide scales. For this purpose, the output data of the regional climate model HIRHAM5, forced by the ERA-Interim reanalysis product (Dee et al.,

2011), are employed as climatic forcing. For the GrIS the mean of the precipitation and surface temperature output between 1980 and 2014 with a spatial resolution of $0.05°$ or $\sim 5\,\mathrm{km}$ are used (Langen et al., 2017; Mottram et al., 2017). For the AIS the mean surface mass balance and $10\,\mathrm{m}$ temperature output between 1980 and 2017 are used (Hansen et al., 2021), which have a resolution of $0.11°$ or $\sim 12.5\,\mathrm{km}$.

In the ice sheet wide studies, new surface layers are formed with a density of $315\,\mathrm{kg\,m^{-3}}$, following Fausto et al. (2018).

Although there will be a high variability of the surface density over the whole GrIS and AIS, the potential bias can be neglected in our studies, as the main interest lies in the identification of the relative changes of the firn properties by strain softening, rather than in absolute values. The residual strain rate in the ice sheet experiments is set to $-2 \times 10^{-4}\,\mathrm{yr^{-1}}$, which is a good approximation of the vertical strain rate when compared to ice sheet and ice rise observations (Zumberge et al., 2002; Elsberg et al., 2004; Gillet-Chaulet et al., 2011).

## 375 5 Results and Discussion

### 5.1 Sensitivity test

In order to understand how strain softening behaves under different dynamic conditions a sensitivity test is conducted. For the climatic conditions present at the EGRIP site, the firn density and age are modeled for a range of effective horizontal strain rates between $\dot{\varepsilon}_{\mathrm{E,H}} = 0$ and $\dot{\varepsilon}_{\mathrm{E,H}} = 7 \times 10^{-3}\,\mathrm{yr^{-1}}$.

The strain dependent age profile of the firn is shown in Fig. 2a. The black contour lines additionally indicate the transition between first and second firn stage at the critical density of $550\,\mathrm{kg\,m^{-3}}$ and the firn-ice transition at the BCO density of $830\,\mathrm{kg\,m^{-3}}$. The depth of the critical density remains unaffected, as the strain enhancement model is only active in the second firn stage. Below the critical density the densification is accelerated as the effective horizontal strain rate is increased and the firn-ice transition occurs at shallower depths. This reduction is strongest at low $\dot{\varepsilon}_{\mathrm{E,H}}$ and steadily gets weaker when strain

rates rise. For example the thinning at an effective strain rate of about $1.2 \times 10^{-3}\,\mathrm{yr^{-1}}$ is already half as big as in the case of $\dot{\varepsilon}_{\mathrm{E,H}} = 7 \times 10^{-3}\,\mathrm{yr^{-1}}$. Hence, the sensitivity of firn densification to strain softening is greatest at low strain rates.

The picture is similar for the sensitivity of the firn age at a certain depth and the BCO age, represented in Fig. 2a by the age of the firn at the BCO density line. Again both quantities are most sensitive to the effect of strain softening at low strain rates. The shift of the firn age at a given depth is however rather weak with a decrease of the firn age by up to $10\,\%$ for

$\dot{\varepsilon}_{\mathrm{E,H}} = 7 \times 10^{-3}$ yr$^{-1}$. The BCO age in contrast is strongly affected by the horizontal strain rates with a decrease of $50\,\%$ at the highest strain rate values in the test.

In summary, both the BCO depth as well as the BCO age are strongly affected by high strain rates, but even low strain rates of less than $1 \times 10^{-3}$ yr$^{-1}$ do affect the firn properties considerably. Especially this fact highlights the need for the tuning bias correction. Due to the initially strong effect of strain softening, an implicit contribution of $\dot{\varepsilon}_{\mathrm{cor}} = 4.5 \times 10^{-4}$ yr$^{-1}$, as it is

expected for the HL model, already matters.

## 5.2   Comparison with firn cores

The firn density profiles at the sites of the NEGIS and the EGRIP S5 2019 firn cores are modeled with the HL model by first considering no strain and subsequently activating the modules for horizontal divergence (not shown), strain softening and the tuning bias correction (TBC).

In Fig. 2b the results for the NEGIS firn core, drilled at the site of the EGRIP station, are shown and compared to the data. The mean effective horizontal strain rate over the firn age at this site is indicated by the corresponding vertical line in 2a at $\overline{\dot{\varepsilon}_{\mathrm{E,h}}} = 0.42 \times 10^{-3}$ yr$^{-1}$. As it is small, the differences between the various model setups are also small. The no-strain model already reproduces the density data with good agreement. Horizontal divergence only contributes to a reduction of the firn thickness by $1$ m and strain softening thins the firn by additional $7$ m.

As the no-strain model already matches the data, the strain softening enhancement leads to an underestimation of the firn thickness. But by this fact, the tuning bias becomes apparent and the underestimation should not be attributed to the strain softening model, but to the underlying HL model. $\overline{\dot{\varepsilon}_{\mathrm{E,h}}}$ almost matches $\dot{\varepsilon}_{\mathrm{cor}}$. Accordingly, the firn cores used for tuning the HL model have on average experienced as much stain as the firn at EGRIP and the corresponding strain softening contribution can be expected to be already captured by the HL model. As a consequence, this contribution is considered twice, when our strain

softening enhancement model is applied. In order to only consider it once, the previously introduced tuning bias correction has to be added. It increases the firn thickness again by around $7$ m and thereby suppresses the effect of the strain softening model.

Although the shear margin firn density data of the S5 2019 firn core are not available, the corresponding modeled firn density profiles are shown in Fig. 2c as an example for a high-strain site with $\overline{\dot{\varepsilon}_{\mathrm{E,h}}} = 2.9 \times 10^{-3}$ yr$^{-1}$. The modeled density profile for the cases of no strain resembles the profiles at the EGRIP site. Horizontal divergence again only has a minor effect on the firn

profile and is not shown. Strain softening causes a significant reduction of the firn thickness by $30$ m and thereby causes the kink at the critical density to disappear. Yet, the tuning bias in the HL model is expected to be as big as on the EGRIP site and the correction for it increases the firn thickness again by $7$ m.

As before, these numbers have to be understood in the way that at the S5 site strain softening causes firn thinning of $30$ m, but $7$ m thereof are already captured by the HL model, giving a total reduction $23$ m when the strain softening enhancement

and tuning bias correction are applied.

## 5.3 Validation with NEGIS firn density cross section

As a next step, the firn densities recorded by Riverman et al. (2019) along a cross section of NEGIS are reproduced. The original data are shown in Fig. 3a with the firn-ice transition being indicated by the white contour line. In this profile the increased firn density within the shear margins of NEGIS can be seen, causing a reduction of the firn thickness by 20 to 30 m.

In Fig. 3b the density profile is modeled with the tuning bias corrected strain softening model at the locations of the individual seismic survey sites as shown in Fig. 1a, according to the forcing parameters displayed in Fig. 3c, which are the accumulation rate $\dot{b}$ and the mean effective horizontal strain rate over the firn age $\overline{\dot{\varepsilon}_{E,H}}$. For the temperature, a constant value of $-29.9°$C was assumed. The solid white line indicates the BCO depth of the corrected strain softening model. Additionally, it is also displayed for the model runs with no strain, horizontal divergence and the uncorrected strain softening model.

In the case of no strain, the BCO line represents the impact of the accumulation rate variability alone. It shows that the observed density peaks in the shear margins cannot be attributed to the accumulation pattern. In contradistinction to the observed lower firn thickness, the higher snow accumulation in the shear margin would even promote an increase of the firn thickness.

Similar conclusions need to be drawn for the effect of horizontal divergence, as the corresponding BCO line only differs slightly from the case of no strain. Horizontal divergence merely affects the firn thickness by a few meters, whereat no clear trend for a firn thinning can be seen, but instead, firn thickness also increases where velocities are actually converging as it is for example the case at the S5 2019 firn core site, indicated by the left vertical line. If on a flat ice sheet velocities diverge at one place, they tend to converge elsewhere. Therefore, horizontal divergence cannot explain a pure firn thinning pattern but always results in an increase of firn thickness nearby. Consequently, horizontal divergence cannot explain the reduced firn thickness in the shear margins of ice streams.

The increased shear margin firn density with the respective lowering of the BCO depth can only be reproduced, when strain softening is included in the model. In this case the extent of the density peaks can be reproduced well, which validates the model and supports the idea that the enhanced firn densification rates in high-strain environments are caused by strain softening.

Comparing the strain softening cases, modeled with and without applying the tuning bias correction, it becomes notable, that even the small contribution of $\dot{\varepsilon}_{cor}$ can considerably alter the modeled firn thickness by around 7 m. The impact of the correction only varies little over the cross section as the climatic forcing also varies little. Therefore, the correction basically reduces the initial densification rate of the HL model equally over the whole section before the strain softening enhancement is applied.

Outside the ice stream, where effective horizontal strain rate forcing is even lower than $\dot{\varepsilon}_{cor}$, the firn thickness is increased. Thereby, the fit of the tuning bias corrected strain softening model is not only better in the high-strain shear margins, but also in the low-strain sections of the profile. This indicates that the tuning bias correction and the sensitivity of the model are accurate. It especially supports our interpretation that some fixed implicit strain softening contribution is captured in the Herron-Langway model – and potentially also in other empirical climate-forced firn models – and that this contribution will lead to a mismatch of the model, whenever the site-specific strain rate forcing differs from the forcing that caused the tuning bias.

The main difference between data and model in Fig. 3b lies in an apparent shift of the modeled firn density profile of $2\,\mathrm{km}$ towards south-east. No processing error that would explain such a shift could be identified, neither in the model nor in the data. Instead, the shift could point towards a potential movement of the ice stream position over time, which would not be represented in the model, because the past horizontal strain rates are inferred from present-day surface velocities, whereby the assumption of steady state conditions is implicitly made.

## 5.4 Firn properties along NEGIS

In Fig. 4a the modeled change of the firn thickness is compared with the elevation of the ice sheet surface along the seismic survey line. The shear margin troughs resemble the modeled reduction of the firn thickness with regard to depth, extent and location, suggesting that they are formed by a collapse of the firn layer due to strain softening. However, as the agreement is not perfect, other factors, such as upstream effects, accumulation variations and the sub-glacial topography, must also alter the surface elevation and the structure of the shear margins troughs.

Previously, the expected depth of these troughs was estimated by integrating the vertical strain rate caused by horizontal divergence along the flow line, giving a total strain of $-0.1$ which translates into a trough depth of $200$ to $300\,\mathrm{m}$ (Fahnestock et al., 2001). Our results on the contrary indicate that horizontal divergence in the firn does not contribute to the trough formation. Although our study focuses on the firn layer, this conclusion can be extended to the ice layer below. Because the firn thinning matches the depression of the surface topography, the ice layer itself has an approximately flat surface, suggesting that no thinning is occurring within it. Instead, it indicates that the pattern of horizontal strain rates must be changing with depth. This in turn might have an impact on strain softening, which is not captured in the presented model, as we assume that horizontal strain rates are constant with depth. Hence, the internal dynamics at the ice stream shear margins and the impact of the depth-variable strain rates need to be investigated in future studies.

The BCO age along the profile is shown in Fig. 4b. While the firn reaches an age of up to $400\,\mathrm{yr}$ in the center of the ice stream, the firn-ice transition in the shear margin already takes place after about $200\,\mathrm{yr}$. This means that in the shear margins the densification rate is effectively doubled by strain softening. The BCO age is reduced by $50\,\%$ over a distance of only $5\,\mathrm{km}$, whereas the climatic forcing on firn air processes can be expected to vary little over such short distances. For this reason, we suggest to exploit the shear margins as a natural laboratory for firn air studies, in order to better constrain characteristic parameters of firn air processes.

## 5.5 Implications for ice core dating

The gas enclosed in bubbles at the lock-in depth is younger than the surrounding ice. This introduces a difference between the age of the ice matrix and the gas in an ice core ($\Delta$age). Accurate models of densification are thus important for synchronizing ice core records. Precise relative timing is necessary to understand the sequence of events and the physical mechanisms behind past changes in climate (Pedro et al., 2012). For example Buizert et al. (2015) find that abrupt Greenland warming events lead corresponding Antarctic cooling onsets by $218 \pm 92\,\mathrm{yr}$. This conclusion hinges on the estimated $\Delta$age at the WAIS Divide

ice core. In contrast, Svensson et al. (2020) find a $122 \pm 24\,\mathrm{yr}$ lag between Greenland and Antarctic ice core records using a volcanic synchronization that does not rely on densification processes.

To gauge the potential impact of strain softening at this site, we test the sensitivity of the firn age to an exemplary effective strain rate of $1 \times 10^{-3}\,\mathrm{yr}^{-1}$ using WAIS Divide climate conditions. Therefore, the strain softening model without the tuning bias correction is applied in order to obtain the total contribution by strain softening. For Holocene climate we find that an effective strain rate of $1 \times 10^{-3}\,\mathrm{yr}^{-1}$ reduces the BCO age, which we use as an approximation of $\Delta$age, by $23\,\%$ (from 309 to $238\,\mathrm{yr}$) and BCO depth by $20\,\%$. For Last Glacial Maximum conditions ($-41\,°\mathrm{C}$ and $0.1\,\mathrm{m\,yr}^{-1}$) the BCO age is reduced by $33\%$ or $209\,\mathrm{yr}$ and BCO depth by $29\,\%$.

Accordingly, strain softening can affect the $\Delta$age in two ways: On the one hand, it is reduced when strain rates rise. Accordingly, if the flow pattern has changed in the past, strain rates might have been different. At sites where this is the case, tuning a firn densification model to the local Holocene conditions can still induce a bias in the $\Delta$age estimates inferred for the past. On the other hand, the impact of strain softening depends on the climatic forcing at the site itself and can alter over time even if the effective horizontal strain rate remains constant. In our sensitivity test for the WAIS Divide site, the BCO age reduction by a moderate strain rate forcing decreased from 209 to $71\,\mathrm{yr}$ between the Last Glacial Maximum and the Holocene.

The decrease of the age difference by strain softening, as well as its variability under stable dynamic conditions, is therefore on the order of the observed time lag between Greenland and Antarctic ice core records and needs to be considered for synchronizing them by the methane ($CH_4$) record. The shift of $\Delta$age can thereby be positive or negative. A climate forced model which has been calibrated to the locally observed density profile will implicitly account for the local present-day horizontal strain rate. Depending on whether past horizontal strain rates were greater or smaller than at present, the strain softening correction to such a model can also work in either direction. We stress that our WAIS Divide modeling only constitutes a sensitivity test and should not be taken as an error estimate of existing firn models which are tightly constrained by the present-day firn density profile and $\delta^{15}\mathrm{N}$ data (Buizert et al., 2015).

In classical climate-forced densification models the densification rate and thus $\Delta$age is almost entirely determined by surface temperature and accumulation rate. Buizert et al. (2021) exploit this to infer past surface temperature from estimates of $\Delta$age and accumulation rate. However, our modeling shows that horizontal strain leads to enhanced densification rates and should also be taken into account. The large-scale ice flow could have changed over time, which complicates the modeling of past densification rates. Indeed, accumulation changes must be accompanied by changes in ice flow speeds, and thus strain rates, in order to maintain flux balance. While the results by Buizert et al. (2021) agree with the temperature inferred from borehole thermometry, our observations highlight that in regions with a strong dynamical history strain softening needs to be considered. The opposition of these observations can however also be used to infer an upper limit for the past strain rates, if the firn densification-derived temperatures are backed up by an independent method.

## 5.6  Firn properties of the polar ice sheets

Finally, the ice sheet wide impact of strain softening on firn densification is studied. For this purpose, a range of steady-state firn density profiles are modeled with the HL model and the strain softening extension, but without the tuning bias correction

being applied, to create a data grid that can be used to obtain the approximate change of the BCO depth and BCO age by strain softening at every point on the ice sheet in a computationally efficient manner by interpolation. These profiles encompass various combinations of forcing parameters that cover the range of climatic conditions and effective horizontal strain rates that are present over the GrIS and the AIS according to the multi-year average of the HIRHAM5 output data and the satellite-based velocity field products.

For Greenland, temperature was altered between $-29°\text{C}$ and $-17°\text{C}$ in steps of $2°\text{C}$, accumulation rate was logarithmically increased in 7 steps from $75\,\text{mm}\,\text{yr}^{-1}$ to $1\,\text{m}\,\text{yr}^{-1}$ and the effective horizontal strain rate was increased in steps of $1\times10^{-3}\ \text{yr}^{-1}$ from 0 to $7\times10^{-3}\ \text{yr}^{-1}$, giving 392 different combinations of forcing parameters in total. For Antarctica temperature was linearly increased from $-60°\text{C}$ to $-10°\text{C}$ in 6 steps, accumulation rate was again logarithmically increased in 9 steps from $5\,\text{mm}\,\text{yr}^{-1}$ to $1\,\text{m}\,\text{yr}^{-1}$ and the effective horizontal strain rate was increased up to $10\times10^{-3}\ \text{yr}^{-1}$ with the same spacing as before. Which gives 594 different combinations for the AIS.

Locations with warmer temperatures and lower accumulation rates than given by these input ranges were not modeled and for the GrIS also places with an average annual melt of more than $1\ \text{mm}$ were excluded. This was done because in the ablation zone of the polar ice sheets additional processes like melt-water percolation and refreezing contribute to firn densification, which will not be enhanced by strain softening. Drawing conclusions about how strain softening affects the general firn densification process in these areas is therefore not easily possible. To not overestimate the contribution of strain softening in these areas, we decided to restrict our studies to the dry zone of the polar ice sheets by introducing the above restrictions.

The BCO depth and BCO age for the input forcing were then determined from the modeled steady-state firn profiles. Local firn properties at every point on the ice sheet were obtained by linear interpolation of the local climatic forcing to the parameter grid. With this approach the ice sheet wide contribution of strain softening to firn densification can be studied by comparing the interpolated firn thickness in the cases of no strain to the case when the uncorrected strain softening model is employed.

Figure 5 shows for both polar ice sheets the firn thickness when strain softening is considered (a & d), as well as the absolute (b & e) and the relative (c & f) change of the firn thickness that is caused by strain softening. The figure illustrates that strain softening on both ice sheets significantly reduces the firn thickness in the shear margins of ice streams and the onset regions of the outlet glaciers. A considerable contribution is also observed over the fast-flowing Antarctic ice shelves. When looking at the relative firn thinning of the AIS in Fig. 5f, it can moreover be noted that even in the interior of the East Antarctic ice sheet (EAIS) strain softening enhances firn densification by up to $10\,\%$, despite low flow velocities and therefore also low horizontal strain rates being present. This unexpected observation can be explained by very low temperatures and accumulation rates occurring in this region, which give rise to extremely low firn densification rates with a BCO age on the order of $10^3$ yr and in consequence enable strain softening to have a relatively strong impact by being active over a long period.

However, strain rates derived from remotely sensed velocities are sensitive to the degree of smoothing applied. Spatially uncorrelated velocity noise will lead to a positive bias in the effective strain rate. Smoothing reduces the noise amplitude and will act to lessen this bias. Unfortunately, smoothing will also blur the true strain rate signal leading to a negative bias in the effective strain rate in high-strain regions. The degree of smoothing is therefore a compromise between reducing noise, and not degrading the signal too much. In this paper, we have chosen the degree of smoothing necessary to remove obvious

artifacts. However, in the interior regions with very slow flow the true strain rates may be so small that even a tiny remaining noise amplitude can still be a substantial component of the estimated effective strain rate. This is an important caveat when interpreting the continental scale maps in Fig. 5. For example the observed strain softening contribution in the interior of the EAIS could therefore also be caused by an overestimation of the strain rates due to remaining noise.

The velocity maps moreover contain spots in the polar hole, where data are missing and where it therefore is not possible to model the effect of strain softening. In combination with the difficulties arising from processing artifacts and noise, this highlights the need for high-quality velocity and strain rate data from remote sensing for modeling the effect of strain softening.

    Nonetheless, we see that the strain softening contribution to firn densification matters over wide areas of the ice sheets. This however does not mean that existing firn densification models, which do not consider strain softening, generally overestimate

the firn thicknesses, but rather that these models will likely include some contribution of strain softening in the form of a tuning bias, as previously discussed. To highlight this effect, Fig. 5 additionally shows the locations of the firn cores that were used for empirically tuning the Herron-Langway model. At least at some of these sites, especially on the AIS, firn thickness is appreciably affected by strain softening, which supports our interpretation that the HL model will capture this thinning to some extent.

As we have seen in Fig. 2b, the HL model without any adaptations fits well at the EGRIP site, but it does for the wrong reason because only by chance the strain rate forcing at this site matches the implicit forcing in the HL model. At sites where the effective horizontal strain rate is different, the fit of the HL model is lower. Consequently, strain softening is potentially the reason for why often site-specific recalibration of firn models is needed and why the firn properties over the whole GrIS cannot sufficiently be modeled by only considering climatic forcing, as observed by Simonsen et al. (2013). Imagining that the

magnitude of the tuning bias differs between various firn models, strain softening can in turn also contribute to the mismatch between different firn models, as seen by (Lundin et al., 2017).

    To take account of this implicit contribution, we introduced the tuning bias correction, which is however only a first-order estimate of the implicit strain softening contribution. We want to stress that it is required to consider all three input parameters – i.e. the accumulation rate, the temperature and the effective horizontal strain rate – during the empirical tuning of a firn model

to represent all of them accurately and to capture the sensitivity of firn densification to the three forcing parameters correctly. We leave this to future studies.

## 6   Conclusions

We have developed an extension for firn densification models that is capable of correcting the densification rate of any climate-forced firn model for the effect of strain softening. Employing this model, it was studied how strain softening affects firn

densification on local and ice sheet wide scales.

    We found that the sensitivity of firn densification to strain softening is highest at low strain rates and that therefore even low strain rates can affect the firn thickness considerably in areas where forcing by accumulation rate and temperature is weak. In high-strain areas, such as the shear margins of ice streams, a significant acceleration of firn densification by strain softening was

modeled, which is in good agreement with observations of lower firn thickness in these areas. As other potential processes, like
horizontal divergence or a greater accumulation in the shear margin troughs, could not explain this reduction of firn thickness,
our work supports the idea that strain softening is the principal cause. It was further observed that the change of firn thickness
resembles the lowering of the surface elevation in the shear margins, which suggests that the shear margin troughs form because
of a faster settling of the firn due to strain softening.

Strain softening does not only affect the firn thickness but also reduces the age of the firn at the firn-ice transition. According
to our model this can lead to a reduction of the BCO age by around $50\,\%$ over a few kilometers in the shear margin of ice
streams. We therefore suggest to exploit this feature as a natural laboratory in future firn air studies, because the climatic
forcing over such small distances will only vary slightly. Moreover, strain softening can strongly alter the BCO age over time,
even at constant, moderate strain rates. For ice core dating this induces a bias in the BCO age, which previously has not been
taken into account, but is on a considerable order for synchronizing ice core records by $CH_4$.

Finally, we demonstrate that strain softening has a substantial effect on firn densification over wide areas of ice sheets,
and as a consequence that horizontal strain rates should generally be considered in firn densification modeling, because a
restriction to climatic forcing parameters results in a misrepresentation of the latter. Our work therefore suggests that besides
temperature and accumulation rate also the effective horizontal strain rate should be considered as a relevant forcing parameter
in firn densification modeling and that all three parameters should already be considered during the empirical tuning of firn
densification models.

*Code availability.* The code for the CFM model with the extension for strain softening is available at https://github.com/oraschewski/
CommunityFirnModel/tree/Falk.

*Author contributions.* The idea for the strain softening model extension was developed by both authors. F. Oraschewski carried out the model
experiments and wrote the majority of the code and paper, based on his Master's thesis. A. Grinsted acted as a supervisor for the thesis as
well as in the preparation of the manuscript.

*Competing interests.* The authors declare that they have no conflict of interest.

*Acknowledgements.* We acknowledge the support of the Villum Investigator Project IceFlow (grant no. 16572) for this work and the support
of the German Academic Scholarship Foundation to F. Oraschewski. EGRIP is directed and organized by the Centre for Ice and Climate at
the Niels Bohr Institute, University of Copenhagen. It is supported by funding agencies and institutions in Denmark (A. P. Møller Founda-
tion, University of Copenhagen), USA (US National Science Foundation, Office of Polar Programs), Germany (Alfred Wegener Institute,
Helmholtz Centre for Polar and Marine Research), Japan (National Institute of Polar Research and Arctic Challenge for Sustainability), Nor-

way (University of Bergen and Trond Mohn Foundation), Switzerland (Swiss National Science Foundation), France (French Polar Institute Paul-Emile Victor, Institute for Geosciences and Environmental research), Canada (University of Manitoba) and China (Chinese Academy of Sciences and Beijing Normal University). We further acknowledge the Arctic and Climate Research section at the Danish Meteorological Institute for producing and making available their HIRHAM5 model output. Data from the Programme for Monitoring of the Greenland Ice Sheet (PROMICE) were provided by the Geological Survey of Denmark and Greenland (GEUS) at http://www.promice.dk. We also acknowledge the use of the the National Snow and Ice Data Center QGreenland package, the Quantarctica package provided by the Norwegian Polar Institute and the use of data provided by the SCAR Antarctic Digital Database. We thank Kiya Riverman for providing the firn density and accumulation rate data across NEGIS and Helle Astrid Kjær and Paul Vallelonga for providing the NEGIS firn core data. We thank Nicholas Rathmann for discussions on porous flow.

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

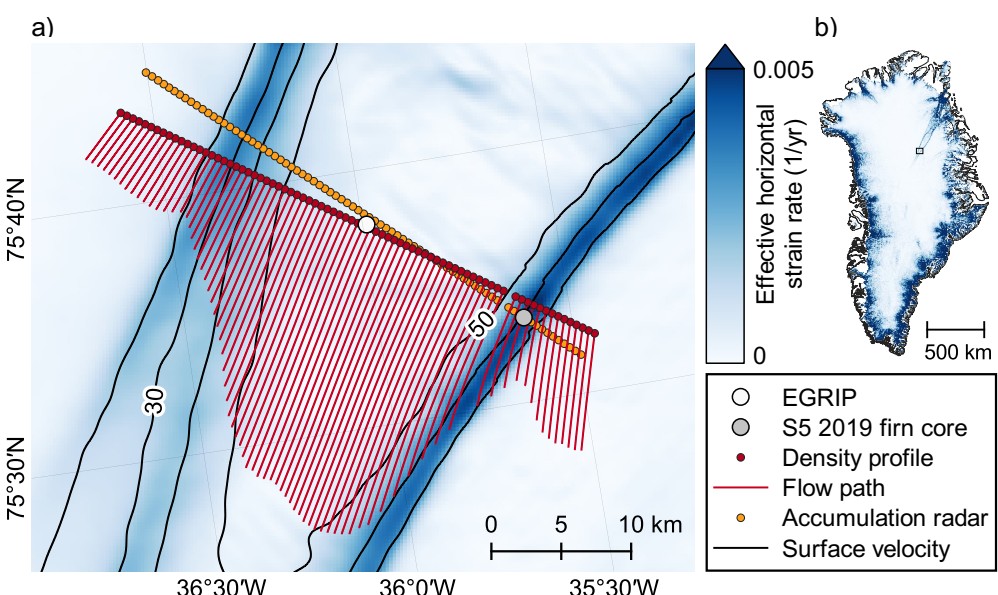

**Figure 1.** (a) Locations of firn surveys conducted at NEGIS around the EGRIP station. The map shows the sites of the NEGIS firn core (at EGRIP) and the S5 2019 firn core. The red dots indicate the coordinates of the active seismic surveying sites that form the density profile by Riverman et al. (2019) and the yellow dots the corresponding closest points on the Operation IceBridge accumulation radar line. The red lines indicate the backtraced flow path over the firn age at the density profile points. In the background, the effective horizontal strain rate and surface velocity contours (in $\mathrm{m\,yr^{-1}}$) are shown. (b) Effective horizontal strain rate over the Greenland ice sheet. The black box outlines the extent of (a).

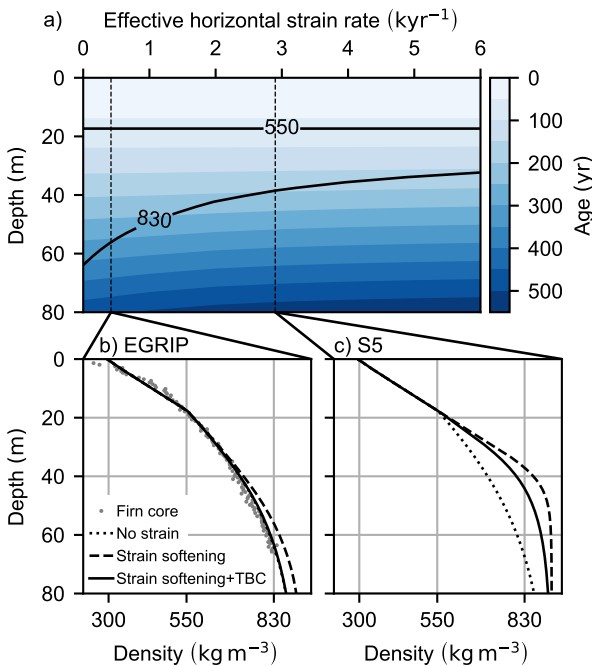

**Figure 2.** (a) Study of the decreasing sensitivity of firn densification to strain softening for increasing effective horizontal strain rates for climatic conditions present at EGRIP. The background shows the age contours and the black lines indicate the critical depth and the depth of the firn-ice transition. Vertical dashed lines indicate the backtraced mean effective horizontal strain rate at the firn core sites in (b) and (c). (b) Firn density data and models of the NEGIS firn core at EGRIP and its model with no strain, strain softening and additionally the tuning bias correction (TBC) being applied. (c) The same model outputs for the S5 2019 shear margin firn core.

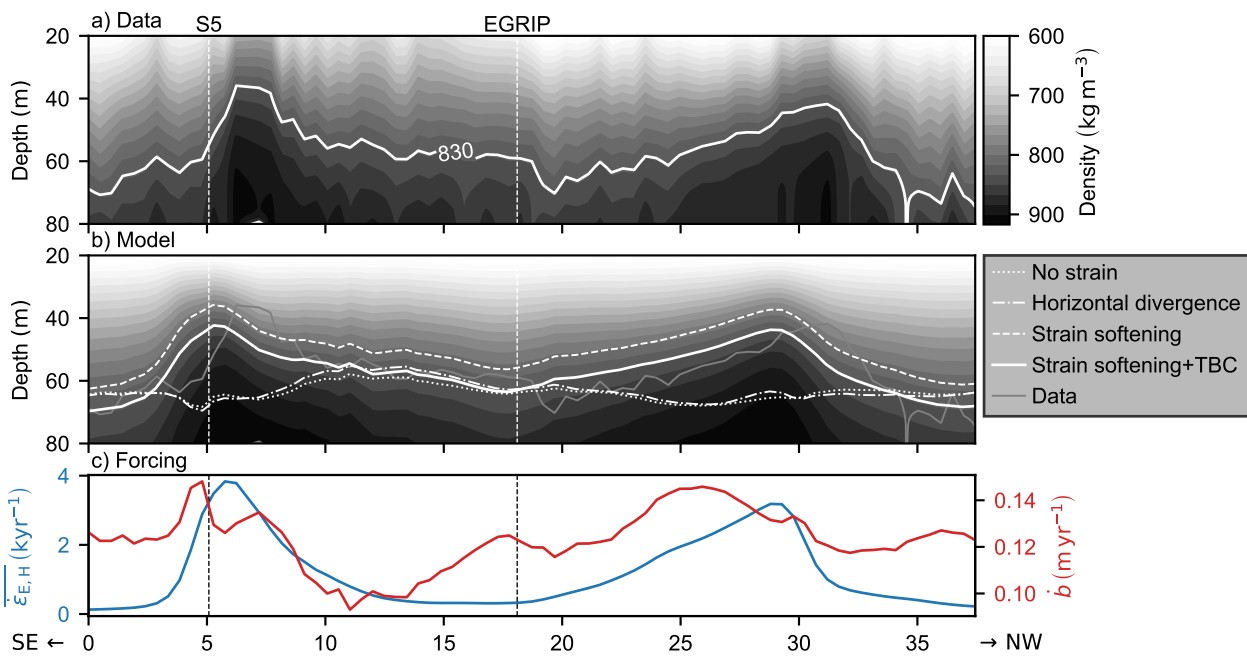

**Figure 3.** (a) Firn density profile along a cross section of NEGIS recorded by Riverman et al. (2019, Fig. 9b). The white contour line indicates the firn-ice transition. (b) Modeled firn density profile for the same location using the tuning bias corrected (TBC) strain softening model, where the white line again indicates the firn-ice transition. The latter is also shown for the cases of no strain, horizontal divergence and the uncorrected strain softening model. (c) Mean effective horizontal strain rate over the firn age and the radar-derived accumulation rate, which are used for forcing the models in (b). The dashed vertical lines indicate the locations of EGRIP and the S5 shear margin firn core site.

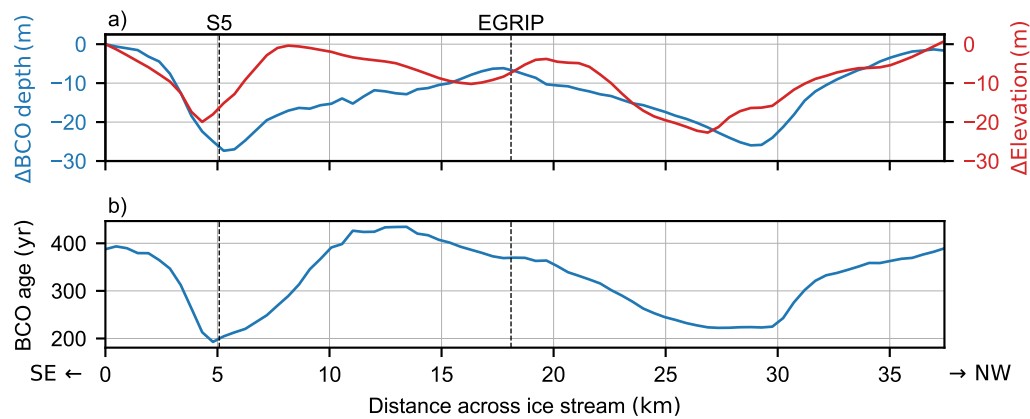

**Figure 4.** Firn properties along the NEGIS density profile, according to the tuning bias corrected strain softening model, as shown in Fig. 3b. (a) The change of the bubble-close off (BCO) depth and the surface elevation with respect to the first data point. (b) Age of the firn at bubble-close off.

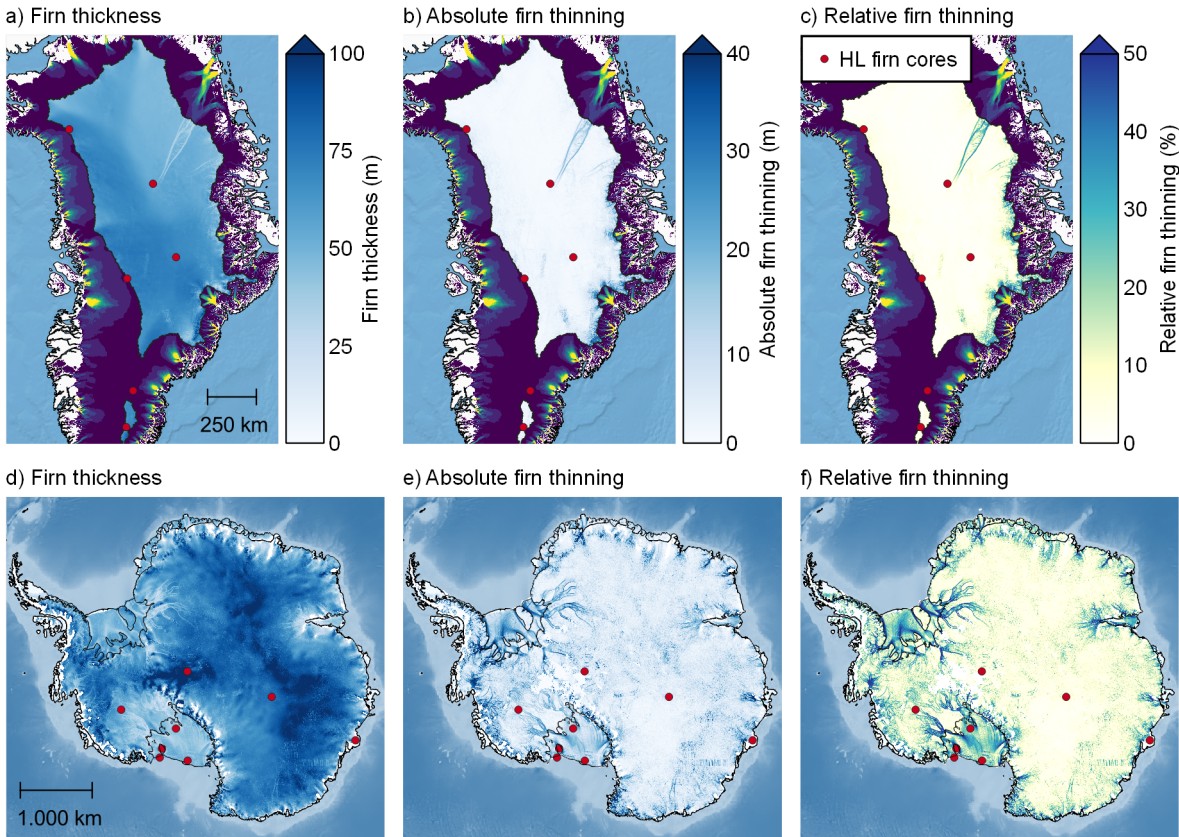

**Figure 5.** Study on the contribution of strain softening to firn densification in terms of the firn thickness over the dry zone of the Greenland ice sheet (GrIS, a-c) and the Antarctic ice sheet (AIS, d-f). (a & d) Modeled total firn thickness when strain softening is considered without the tuning bias correction. (b & e) Absolute firn thinning contribution caused by strain softening. (c & f) Relative firn thinning due to strain softening, which also illustrates by how much the densification process is accelerated by strain softening. Red dots indicate the drill locations of the firn cores that were used for tuning the empirical Herron-Langway model and illustrate that some of these firn cores were considerably affected by strain softening. Outside the dry zone of GrIS the surface velocity is shown with brighter colors indicating faster flow.