# Peer review of "Modeling enhanced firn densification due to strain softening"

_The Cryosphere, 2021_

## Referee Comment (RC1)

Review **tc-2021-240** *"Modeling enhanced firn densification due to strain softening"* by Oraschewski and Grinsted

This paper proposes a method to account for strain softening in firn densification models. Most of the densification models used to predict the evolution of density with depth in polar ice-sheets are based on the assumption that the horizontal flow can be neglected, assuming an infinitely large and flat firn. Doing so, the density is only function of depth (1d model). Such assumption might hold at places such as the central parts of these ice-sheets (domes, ridges), but is certainly not anymore valid closer to margins. The proposed method account for strain softening by including the horizontal strain-rates in the effective strain-rate. Horizontal strain-rates ($\dot\epsilon_{xx}$, $\dot\epsilon_{yy}$ and $\dot\epsilon_{xy}$) are assumed vertically uniform and estimated from surface velocity. The overall paper is well written and the figures are of good quality. I have nevertheless a number of criticisms on the model itself and on some hidden assumptions used to derive the model.

Most of the derivations rely on the assumption regarding the **form of the strain-rate tensor** (Eq. (4)). It is assumed that the shear components $\dot\epsilon_{xz}$ and $\dot\epsilon_{yz}$ are negligible in comparison to $\dot\epsilon_{xx}$, $\dot\epsilon_{yy}$ and $\dot\epsilon_{xy}$. This assumption might be true for an incompressible ice column if one looks only at the top of the ice column as shear deformation will mainly concentrate close to the bedrock. But this is not anymore true if the upper part of the column is composed by a compressible material like firn (see as an example the vertical profil of horizontal velocity in Fig. 9.16 of Greve and Blatter (2009), which indicates larger $\dot\epsilon_{xz}$ close to the surface than at the base). At least, this assumption should be discussed!

There are a number of **implicit assumptions** in the derivation of the model that are not clearly stated. The unknown strain-rate components $\dot\epsilon_{xx}$, $\dot\epsilon_{yy}$ and $\dot\epsilon_{xy}$ are derived from the surface velocity, which implicitly assumes that they are depth uniform. This assumption should be clearly stated (much before line 314) and discussed more deeply. How much the fact that the firn is compressible will contradict this assumption, as for $\dot\epsilon_{xz}$ and $\dot\epsilon_{yz}$? I think that there is an other implicit assumption to derive Eq. (5): the firn particules flow only vertically, which then contradict the initial assumption that $\dot\epsilon_{xx}$, $\dot\epsilon_{yy}$ and $\dot\epsilon_{xy}$ are non zero. In fact, one should follow the particules trajectory and integrate density along these trajectories from the surface down to a given depth. In other words, it is assumed that the flow is not anymore only along a vertical flow line, but some equations are still derived using the opposite assumption.

Eq. (15) is used to derive $r_V$, but I am wondering how $\dot\epsilon_{zz,c}$ is estimated? This is not clearly stated anywhere.

**Minor remarks:**

- line 19: that that

- line 57: suggested by Alley and Bentley (1988) to accelerate

- line 70: it is not because it has never been applied to polar ice-sheets that it cannot! By the way, the density functions $a$ and $b$ in the model developed by Gagliardini and Meyssonnier (1997) are calibrated using the density profile of Site 2 in Greenland. I am wondering what should we conclude from this paragraph regarding the approach of Gagliardini and Meyssonnier (1997)? Be more specific.

- line 103: it is not clear here if you are speaking about your model or previous models that only consider vertical compaction? The proposed constitutive relation only hold in the later case.

- line 107: this sentence about incompressibility is not clear. Firn is compressible, so there is no need of an incompressibility assumption. For a compressible material, the trace of the strain rate tensor is not null anymore, such that the two constitutive relations should be set between (i) deviatoric stress and deviatoric strain rate and (ii) trace of the strain rate (change in volume) and isotropic pressure (trace of the Cauchy stress tensor). See equations (9.53) and (9.54) in Greve and Blatter (2009) for example. Using the formalism proposed by Gagliardini and Meyssonnier (1997), one would not obtain Eqs. (7) and (8). There are certainly some other implicit assumptions beyond these derivations that should be stated more clearly (replacing a deviatoric stress by a Cauchy stress should be justified and motivated).

- line 144: as Oraschewski (2020) is a Master thesis, I think it would be better to include these results in an Appendix. Also, should write "derived in Oraschewski (2020).". Same line 158.

- line 144: it should be explained here how the different strain-rates in Eq. (16) will be estimated (from surface velocity for $\dot{\epsilon}_{xx}$, $\dot{\epsilon}_{yy}$ and $\dot{\epsilon}_{xy}$ and some explanations are needed for $\dot{\epsilon}_{zz,c}$)

- line 179: the fact that $\dot{\epsilon}_{xx}$, $\dot{\epsilon}_{yy}$ and $\dot{\epsilon}_{xy}$ are estimated from surface velocity and thus assumed depth uniform should be mentioned much earlier

- line 210: it is not clear if you account for temperature evolution with depth?

- line 237: sensitivity to what?

- line 271: NEGIS by Riverman et al. (2019) are reproduced.

- line 286: I am not sure that this statement is true? Horizontal velocity from a circular ice cap are always divergent?

- line 314: strange to discuss an assumption that is not even mentioned before

- line 330 and below: units for year is yr. Some time, units are not italicized, some time they are (and should not be, as stated by TC rules). Check this over all the manuscript.

- line 356: over which period are you lookin for in term of climate forcing?

---

## Author Response (AR1)

**Author's point-by-point response to the review of tc-2021-240**

Dear Editor Florent Domine,

we are very thankful for the reviews that we received for this manuscript. They showed that the motivation of, and the assumptions in our model as well as important implications of our results did not become clear. To fix these issues, we have completely revised the theory in section 2, specifically state the assumptions and have included the subsection 3.1 on the assumption of depth uniformity. The correction of the implicit contribution in classical firn models is now introduced in section 2.5 and denoted as a tuning bias correction, to discuss this important aspect in more detail and avoid confusion. Accordingly, we have reformulated mayor parts of our discussions in section 5 to be more clear about how the implication of our results need to be understood.

Best regards,
Falk and Aslak

**Response to review 1 on tc-2021-240**

Thank you for taking your time to thoroughly review this paper. Your comments were of great value for improving the communication of assumptions and the theoretical basis of the paper.
* * *
**General remarks**

This paper proposes a method to account for strain softening in firn densification models. Most of the densification models used to predict the evolution of density with depth in polar ice-sheets are based on the assumption that the horizontal flow can be neglected, assuming an infinitely large and flat firn. Doing so, the density is only function of depth (1d model). Such assumption might hold at places such as the central parts of these ice-sheets (domes, ridges), but is certainly not anymore valid closer to margins. The proposed method account for strain softening by including the horizontal strain-rates in the effective strain-rate. Horizontal strain-rates ($\dot{\varepsilon}_{xx}$, $\dot{\varepsilon}_{yy}$ and $\dot{\varepsilon}_{xy}$) are assumed vertically uniform and estimated from surface velocity. The overall paper is well written and the figures are of good quality. I have nevertheless a number of criticisms on the model itself and on some hidden assumptions used to derive the model.

> **Author response**: Thank you again for your comments. After reading your comments it has become clear that we must make a more clear separation between the theory of the strain softening correction and our application of that correction. It is only our application that relies on horizontal strain rates being uniform with depth, and $\dot{\varepsilon}_{xz}$ being zero. The correction itself will also work in the more general case. We have revised the manuscript to clarify this point.

> **Review**:
> To address these general remarks, we have completely revised section 2 and section 3 to better motivate our theory and assumptions and to have a clear separation between theory and application.

Most of the derivations rely on the assumption regarding the **form of the strain-rate tensor** (Eq. (4)). It is assumed that the shear components $\dot{\varepsilon}_{xz}$ and $\dot{\varepsilon}_{yz}$ are negligible in comparison to $\dot{\varepsilon}_{xx}$, $\dot{\varepsilon}_{yy}$ and $\dot{\varepsilon}_{xy}$. This assumption might be true for an incompressible ice column if one looks only at the top of the ice column as shear deformation will mainly concentrate close to the bedrock. But this is not anymore true if the upper part of the column is composed by a compressible material like firn (see as an example the vertical profil of horizontal velocity in Fig. 9.16 of Greve and Blatter (2009), which indicates larger $\dot{\varepsilon}_{xz}$ close to the surface than at the base). At least, this assumption should be discussed!

> **Author response**: We originally assumed that horizontal strain rates were uniform with depth, and $\dot{\varepsilon}_{xz}$ being zero. However, these are not actually necessary from a theoretical point of view. So, we have revised the theory sections of the manuscript to clarify this. We still use these assumptions when we apply it to EastGRIP and the ice sheets. We have added some additional justification for why this is a reasonable assumption in the interior of the ice sheets (we refer to borehole deformation measurements).

In this context we note that Fig. 9.16 in Greve and Blatter (2009) shows the deformation of a borehole from an alpine glacier. At such glaciers the surface slope, flow velocity and the velocity gradient in vertical direction and thus the vertical shear rate are much higher than at the ice sheets. The main application area of our model extension however are the polar ice sheets and ice shelves rather than alpine glaciers, where full-stokes modeling with the approach by Gagliardini and Meyssonnier (1997) model is more suitable to capture the complex geometry of the glacier.

Moreover, Fig 9.16 particularly indicates larger $\dot{\varepsilon}_{xz}$ near the surface, but we only apply the strain softening model to stage 2 of the firn densification. I.e. below the surface layers, where dislocation creep becomes dominant. Therefore, strong vertical shear rates in the near-surface layers are in any case not captured by our model, and they will not affect the densification rate via strain softening.

**Review**:
We have included subsection 3.1 on the "Assumption of depth uniformity" to address this point and state:

**L259**: "For the application of the model, we assume in this paper that the horizontal velocities in the firn are uniform with depth. While, from a theoretical point of view, this assumption is not needed in the strain softening model, it is required due to the lack of internal velocity data".

There are a number of **implicit assumptions** in the derivation of the model that are not clearly stated. The unknown strain-rate components $\dot{\varepsilon}_{xx}$, $\dot{\varepsilon}_{yy}$ and $\dot{\varepsilon}_{xy}$ are derived from the surface velocity, which implicitly assumes that they are depth uniform. This assumption should be clearly stated (much before line 314) and discussed more deeply. How much the fact that the firn is compressible will contradict this assumption, as for $\dot{\varepsilon}_{xz}$ and $\dot{\varepsilon}_{yz}$? I think that there is an other implicit assumption to derive Eq. (5): the firn particules flow only vertically, which then contradict the initial assumption that $\dot{\varepsilon}_{xx}$, $\dot{\varepsilon}_{yy}$ and $\dot{\varepsilon}_{xy}$ are non zero. In fact, one should follow the particles trajectory and integrate density along these trajectories from the surface down to a given depth. In other words, it is assumed that the flow is not anymore only along a vertical flow line, but some equations are still derived using the opposite assumption. Eq. (15) is used to derive $r_{\mathrm{V}}$, but I am wondering how $\dot{\varepsilon}_{zz,\mathrm{c}}$ is estimated? This is not clearly stated anywhere.

**Author response**: We agree, and we have revised our manuscript to be more explicit about our assumptions.

The assumption that horizontal strain rates are uniform with depth is again not necessary from a theoretical point of view, but required for our application of the model as data of how the velocities change with depth are not available. At sites where this information is known from e.g. bore hole deformation measurements, it could easily be taken into account with the model. Nonetheless, this assumption is justified for the polar ice sheets, where the flow velocity is determined by basal sliding (where the ice sheet is not frozen to the bed as it is suspected for ice streams for example) and by internal deformation which is strongest in the bottom part of the ice sheet. The firn in the top layers mainly flows with the movement of the underlying ice. See e.g. Fig. 7 in Gundestrup et al. (1993).

Following the assumption that horizontal velocities do not change with depth, we do follow particle paths in our EastGRIP densification modeling (red lines in Fig1). At every timestep we model the densification rate of the entire profile for a given location by applying the input parameters for that location and then advect the entire profile with the flow (using surface velocities). This process is realized by backtracking the flow path of the firn and recording the horizontal strain rates at each step. These strain rates are then used as input to our model in the reversed order. Temperature and accumulation rate are not backtracked as we assume that the temperature is constant over the region and that the accumulation rate given by radar surveying already is a mean value over the flow path.

$\dot{\varepsilon}_{zz,\mathrm{c}}$ is computed from the densification rate given by the classical (HL) model using Eq. 5. We will ensure that this becomes clear from the manuscript.

**Review**:
To be explicit about the assumption, including new ones made in the revised derivation of the model, we included the paragraph:

**L198**: "The derivation of the strain softening enhancement above relies on a number of assumptions. We have assumed that in the second stage firn densifies by dislocation creep and that this densification is driven by vertical compression. Other densification processes and horizontal compression are therefore assumed to be negligible and strain softening and horizontal divergence are assumed to be independent of each other. We have further assumed that the microscale solid ice deformation is the key process limiting the rate of firn densification. To relate this to the macroscale compaction we assume that the microscale effective strain rate of deformation scales with the macroscale effective strain rate that contains compaction (Eq. 10) and that firn strain rates scale with a characteristic value for the solid ice strain rate (Eq. 11). The scaling constants ($k_1$ and $k_2$) and the characteristic vertical deviatoric stress of the rate-limiting grains ($\tau_{\mathrm{ice},zz}$) are assumed to have no directional dependence but can depend on density, microstructure, temperature, and load. From these assumptions follows that the effective viscosities in the constitutive equations of the rigid and the compressible phase in the firn are assumed to be equally affected when additional strain rates soften the firn. In this way, we could obtain a model for the enhancement of firn densification by strain softening that involves zero free parameters."

**Minor remarks**

- line 19: that that

    **Author response**: fixed.

- line 57: suggested by Alley and Bentley (1988) to accelerate

    **Author response**: fixed.

- line 70: it is not because it has never been applied to polar ice-sheets that it cannot! By the way, the density functions a and b in the model developed by Gagliardini and Meyssonnier (1997) are calibrated using the density profile of Site 2 in Greenland. I am wondering what should we conclude from this paragraph regarding the approach of Gagliardini and Meyssonnier (1997)? Be more specific.

> **Author response**: We can only speculate why people have not applied this nice model on a large scale. Our guess is because it is more computationally expensive, is more difficult to implement, and it has not been calibrated and tested on as wide a variety of different conditions as the old-school classical models. The success of the empirical Herron-Langway model in that regard is truly remarkable. We believe that with our strain softening correction the applicable range of these simple models will be even greater. It is also simple to implement as long as you assume horizontal velocities are constant with depth. This is clearly a big improvement, even if there are edge cases where these assumptions are not 100% valid.
>
> We have examined the Gagliardini and Meyssonnier (1997) formulation (GM97) in more depth in our preparation for this response and will adapt the paragraph accordingly. We find that the strain enhancement predicted by this model depends critically on $a/b$ which, unfortunately, is poorly constrained. See our reply to the comment to line 107 for more details..
>
> **Review**:
> **L68**: "Despite the observational evidence, firn densification models used for polar ice sheets do not capture the effect of strain softening. Based on a constitutive equation for the power-law creep of porous media (Duva and Crow, 1994), Gagliardini and Meyssonnier (1997) have developed a **glacier flow model** that inherently considers compaction – and in particular strain softening – of the firn. It is widely used in studies of alpine glaciers (Lüthi and Funk, 2000; Zwinger et al., 2007; Licciulli et al., 2020), but has not been applied to the polar ice sheets. **The reasons are presumably that this approach is computationally more expensive on large scales, more difficult to implement and that the range of conditions, that this model is calibrated and tested on, is not as wide as it is for the classical firn models.**"

- line 103: it is not clear here if you are speaking about your model or previous models that only consider vertical compaction? The proposed constitutive relation only hold in the later case.

> **Author response**: We acknowledge that our presentation of the assumptions and the motivation behind the method was inadequate. We have therefore revised this entire section to make our reasoning more clear, and the inherent assumptions more explicit. See also our responses below.
>
> **Review**:
> We have included subsection 2.1 on the "Constitutive relations for firn" to present and discuss the theory behind the constitutive relations in detail.

- line 107: this sentence about incompressibility is not clear. Firn is compressible, so there is no need of an incompressibility assumption. For a compressible material, the trace of the strain rate tensor is not null anymore, such that the two constitutive relations should be set between (i) deviatoric stress and deviatoric strain rate and (ii) trace of the strain rate (change in volume) and isotropic pressure (trace of the Cauchy stress tensor). See equations (9.53) and (9.54) in Greve and Blatter (2009) for example. Using the formalism proposed by Gagliardini and Meyssonnier (1997), one would not obtain Eqs. (7) and (8). There are certainly some other implicit assumptions beyond these derivations that should be stated more clearly (replacing a deviatoric stress by a Cauchy stress should be justified and motivated).

> **Author response**: We acknowledge that our presentation of the assumptions and the motivation behind the method was inadequate. We have therefore revised this entire section and now argue for our strain softening correction from another angle.
>
> There are two perspectives we could take when deriving the strain enhancement factor. We could consider firn as a compressible material where compaction is related to isotropic pressure (as in GM97), or we could consider firn as a complex geometrical structure of solid ice and consider the implications for the densification. We have tested both approaches, and prefer the solid ice perspective as explained in the following.
>
> We now start from the perspective that firn is a mixture of air and solid ice, and that firn densification is rate limited by the strain rate of the solid ice. We end this revised section with a summary reiteration of our assumptions. We hope that this makes our reasoning more clear, and the underlying assumptions much more explicit.
>
> The model with all of its assumptions of course needs to be tested against data. A key feature of our model is that it has zero free parameters. Even without any tuning it is able to reproduce the amplitude of the BCO contour at EastGRIP. We consider this to be a very strong validation of the model sensitivity.
>
> We have also derived a strain enhancement from the GM97 model (Gagliardini and Meyssonnier, 1997) in preparation for this response[1]. You are correct that the GM97 model results in a strain enhancement with a different functional form. We find that the strain softening enhancement based on the GM97 model depends crucially on the ratio between the $a$ and $b$ empirical parameters. These two empirical parameteres are determined by calibration to the Site2 density profile. Unfortunately, this calibration is ill-posed in exactly such a way that the ratio between $a$ and $b$ is poorly constrained (See the final section in Gagliardini (2012) on a 'Porous Law for snow and firn in Elmer/Ice'). This in turn means that the GM97-based strain enhancement factor is poorly constrained, and that we should be vary of any quantitative predictions using GM97 at present.
>
> In Fig. R1.1 we compare the impact of the strain enhancement corrections obtained from the GM97 model with the data of Riverman et al. (2019) and our simpler model across the NEGIS density profile. Qualitatively GM97 generally agrees with the
* * *
[1] In order to derive the strain softening enhancement from GM97 we had to assume: 1) that pressure is unaffected by horizontal strains; 2) that the incompressible part of the vertical strain rate is $\dot{\varepsilon}_{zz,i} = -(\dot{\varepsilon}_{xx} + \dot{\varepsilon}_{yy})$; and 3) that $\dot{\varepsilon}_{xz}$ and $\dot{\varepsilon}_{yz}$ are neglible.

[Figure]

Figure R1.1: (a) Firn density profile across NEGIS recorded by Riverman et al. (2019, Fig. 9b). The white contour line indicates the firn-ice transition/BCO depth. (b) Modeled firn density profile for the same location using the corrected strain softening model. The contours show the BCO depth for the cases of no strain, horizontal divergence and strain softening model and the GM97 model.

results of our model - It also predicts a shallower firn pack in the shear margins. Quantitatively we can see that the effect is about half as pronounced when we use the strain enhancement derived using the GM97 model. Our model is better able to capture the observed amplitude of the BCO contour.

We note another issue with the GM97 model: It predicts a relatively higher strain softening contribution in areas with strong horizontal divergence (between 7km and 18km in Fig. R1.1). When the enhancement factor were increased to fit the data, the densification rates in areas of high horizontal divergence would be overestimated. Discussing this issue in full depth would exceed the scope of this manuscript, so to put it simple, it is caused by mixed terms that are introduced in the effective strain rate due to the more complex functional form of GM97 and the necessary assumptions. Deriving a GM97-based scaling factor therefore does not only require a refined calibration of $a$ and $b$, but also requires additional theoretical work on how to deal with horizontal divergence as our current approach is incompatible with data.

So to summarize: Our model is simpler, fits reality better, and has zero free parameters. Thereby, it is valuable on its own for studying the impact of strain softening on firn densification and capturing the effect with little effort in good approximation. GM97 is a more complete (and really nice) physical description — but GM97 is also substantially more complex, and key parameters are poorly constrained. Our data comparison indicate that GM97 needs more empirical calibration before it can be widely applied (e.g. to shear margins near EastGRIP).

**Review**:
We have included subsection 2.1 on the "Constitutive relations for firn" to discuss the two constitutive relations. The motivation for assuming that the effective viscosities in the two constitutive relations are taken as one is derived in subsection 2.2 on the "Compaction on the microscale".

- line 144: as Oraschewski (2020) is a Master thesis, I think it would be better to include these results in an Appendix. Also, should write "derived in Oraschewski (2020).". Same line 158.

  **Author response**: There are no results or conclusions in the present manuscript that rely on $n = 3$. So, it is not strictly necessary to include this as supplementary information. We point to the MSc thesis as a service to the reader. We note that the thesis has been archived at thesiscommons.org, has been assigned a doi, and is easy to access publicly. So we do not see any great value of adding the exact same information as a Supplement or an Appendix.

  There are plenty of papers on TC that cite master theses. In our opinion this is good — as long as they are easy to access. It would be a shame if all that work is treated as lost unless it makes it into the peer reviewed literature.

  The derivation of the approximate residual strain rate of $\dot{\varepsilon}_0 \approx -2 \times 10^{-4} \, \mathrm{yr}^{-1}$, as mentioned in line 158, will be added as supplementary material.

  **Review**:
  To show that this only a service to the reader, we write:

  **L197**: "An alternative solution for **the often used case of** $n = 3$ **is given in Oraschewski (2020)**."

  For the residual strain rate, we eventually decided to not obtain it with or rough estimate – and therefore do not need the supplement anymore – but to rely it on observations. We write:

  **L220**: "The residual strain rate can be obtained by flow modeling or measured using strain gauge instruments (Zumberge et al., 2002; Elsberg et al., 2004) or phase-sensitive radar (Gillet-Chaulet et al., 2011; Zeising and Humbert, 2021)."

  **L347**: "For the residual strain rate a value of $\dot{\varepsilon}_0 = -0.7 \times 10^{-4} \, yr^{-1}$ is applied, following observation at EGRIP by Zeising and Humbert (2021)."

  **L358**: "The residual strain rate in the ice sheet experiments is set to $-2 \times 10^{-4} \, yr^{-1}$, which is a good approximation of the vertical strain rate when compared to ice sheet and ice rise observations (Zumberge et al., 2002; Elsberg et al., 2004; Gillet-Chaulet et al., 2011)."

- line 144: it should be explained here how the different strain-rates in Eq. (16) will be estimated (from surface velocity for $\dot{\varepsilon}_{xx}$, $\dot{\varepsilon}_{yy}$ and $\dot{\varepsilon}_{xy}$ and some explanations are needed for $\dot{\varepsilon}_{zz,\mathrm{c}}$)

  **Author response**: We will specify how the components can be obtained when discussing Eq. 14.

  **Review**:
  **L185**: "Existing purely climate driven densification models, such as the classical HL model, provide us with an estimate of the vertical strain rate ($\dot{\varepsilon}_{zz,c}$) under the assumption that all other components of the strain rate tensor are zero (Eq. 2). The remaining components of $r_\mathrm{h}$ can be estimated from surface velocity observations or flow modeling."

- line 179: the fact that $\dot{\varepsilon}_{xx}$, $\dot{\varepsilon}_{yy}$ and $\dot{\varepsilon}_{xy}$ are estimated from surface velocity and thus assumed depth uniform should be mentioned much earlier

  **Author response**: The theory is more generally valid, but for our specific applications of the theory this assumption is necessary. We will address this issue together with the discussion of the strain rate components in Eq. 14.

  **Review**:
  This point is now addressed in the new subsection 3.1 on the "Assumption of depth uniformity".

- line 210: it is not clear if you account for temperature evolution with depth?

  **Author response**: We do not account for temperature evolution with depth, as we force our model with constant temperature.
  Temperature has a clear impact on the densification rate and the CFM framework does allow for detailed temperature evolution in the firn. However, this is particularly important in the surface layers where the temperature differs between seasons. Deeper in the firn the seasonal amplitude in temperature is much more attenuated. In this paper we are only concerned with firn stage 2 where dislocation creep is dominant and thus strain softening is active. In this stage the annual average temperature is adequate and we decided to simply use constant climatic forcing.
  We note that in theory firn temperature can also be affected by internal heat production. Therefore, we also tested whether strain heating has a considerable impact on firn densification, but it proved to be negligible. See Fig. 5.5 in Oraschewski (2020).

  **Review**:
  **L301**: "Temperature evolution will not be modeled in our experiments, as our focus lies on processes happening in the second firn stage, where seasonal temperature variations are dampened by heat conduction."

- line 237: sensitivity to what?

  **Author response**: Will be changed to "sensitivity of firn densification to strain softening".

  **Review**:
  **L372**: "Hence, the sensitivity of firn densification **to strain softening** is greatest at low strain rates."

- line 271: NEGIS by Riverman et al. (2019) are reproduced.

  **Author response**: fixed.

- line 286: I am not sure that this statement is true? Horizontal velocity from a circular ice cap are always divergent?

    **Author response**: We agree that the statement is not generally true. We will weaken it to "If on a flat ice sheet velocities diverge at one place, they tend to converge elsewhere.".

    We note that we have verified that this is true in the surroundings of EastGRIP — incl. the shear margins. See Fig. 4.1g in Oraschewski (2020).

    **Review**:
    **L422**: "If **on a flat ice sheet** velocities diverge at one place, they **tend to** converge elsewhere."

- line 314: strange to discuss an assumption that is not even mentioned before

    **Author response**: Fixed. It is now clearly stated earlier.

    **Review**:
    See subsection 3.1 on the Assumption of depth uniformity.

- line 330 and below: units for year is yr. Some time, units are not italicized, some time they are (and should not be, as stated by TC rules). Check this over all the manuscript.

    **Author response**: fixed.

- line 356: over which period are you lookin for in term of climate forcing?

    **Author response**: We use the average over the entire RCM model period. For the Antarctic this is 1980–2017, and for Greenland it is 1980–2014. We clarify this by writing "according to the multi year average of the HIRHAM5 output".

    **Review**:
    **L510**: "...according to the **multi-year average of the** HIRHAM5 output data and the satellite-based velocity field products."

**Response to review 2 on tc-2021-240.**

Thank you for taking your time to thoroughly review this paper. Your comments were of great value for improving the discussion of the implications of our work and for assuring that the message of the paper becomes clear.
* * *
**General remarks**

The authors present a simple modification that allows for the inclusion of strain in calculating firn densification rates. The model provides a satisfactory fit to available active seismic firn density estimates through the NEGRIS shear zones. The model as formulated further has large implications for firn densification away from high-shear environments, although this aspect of the model is not validated in a meaningful way. The modeling approach is clever and interesting, and overall the paper is well written and illustrated. The paper is suitable for publication in TC after some modifications.

> **Author response**: Thank you. In this paper we focus on the theoretical development of the strain softening correction. This is a correction factor which models how much faster the firn is densifying when subjected to horizontal strain. We develop the theory, make sensitivity tests, validate, and study the implications on continental scales. We do not re-calibrate the particular densification model that we apply the correction to. It is, however, only a correction factor and not a complete densification model. This means that any validation must focus on changes rather than absolute values — e.g. $\Delta\rho$ vs $\rho$ or $\Delta z_{BCO}$ vs $z_{BCO}$.
>
> In our validation we demonstrate that our model can fully explain the shallower firn pack observed in the shear margins of NEGIS. The fit is remarkable considering there is no free parameters in the model at all. This strongly indicates that the physical mechanism is real, and that our model has the correct sensitivity.
>
> The strain softening mechanism will of course also be active at low strain sites. However, applying our correction to low shear sites will not automatically yield better fitting profiles than the classical 1D models, as they have been calibrated against real world profiles from low shear sites. Thus they already implicitly account for some small amount of strain softening in their empirical parameters. For this reason it is not very useful to validate our model against $\rho(z)$ at a low strain site without re-calibrating the underlying firn model. This is a big task in itself and would require a whole database of firn profiles. This is beyond the scope of this paper, where we focus on the theory and implications, but it is certainly something that should be done in future work.
>
> This is also why we frame our continental scale results in terms of changes rather than absolutes.

The model suggests strong impacts on firn densification rates for both high- and low-strain environments. For the former, the authors provide some validation using the NEGIS seismic data. However, for the latter the authors make big claims but do not validate their model in any meaningful way.

**Author response**:

We find that the effect is substantial and important even at low strain rate sites, but we also emphasize that the effect is implicitly included in the empirical calibration of existing models (specifically Herron-Langway). So, when we estimate a 10% reduction in BCO depth due to strain enhancement in the interior of Antarctica, then we are claiming that old-school models like Herron-Langway would have errors of that magnitude. We believe this to be at the root of a misunderstanding concerning just how big our claims are.

We demonstrate that the physical mechanism is real. We also show that the sensitivity of the correction is correct in the high strain rate margins of EastGRIP. All physical reasoning tells us that the same mechanism would also be active at low strain rate sites, but we would expect the effect to be less pronounced.

Our model takes the form of a correction to classical models such as Herron-Langway (HL). For low shear environments the strain softening effect is smaller and can be easily compensated for by slight adjustments to the empirical constants in the classical models. Indeed, the empirical constants in a HL model have been tuned to environments with a small amount of horizontal strain, and so those constants already do account for some average strain softening across those sites (as discussed in the paper). So, for low strain environments we expect the HL to already provide a reasonable fit on average. However, being good on average is not good enough for many applications (e.g. ice core interpretation), and so it is common practice to re-calibrate a new densification model at every site. Our model explains how surface strain rates could possibly explain some of the inter-site variability. We therefore argue that all classical 1d models should be recalibrated with this effect in mind.

Therefore, cores from low shear sites are not a good validation of the correction in our model. We would have to start by adjusting the constants in the Herron-Langway model to compensate for the fact it has not been tuned to a zero shear environment. That would slow the densification of the HL model. Then we would apply our strain softening correction which would speed it up. In our opinion it would simply not be a convincing validation to show that our corrected model fits better than the slowed HL model. We would need at least two profiles from the same site but subjected to different strain rates in order to isolate the effect of the correction.

**Review**:
To address this issue, we have adapted our discussions throughout section 5 and introduce the "Tuning bias correction" already in susbsection 2.5 with the hope that in this way our reasoning becomes more clear.

I request that the authors us firn density data to validate their model, because such data are the only true way to test the model validity. Unfortunately, a firn core from the NEGIS shear zone (EGRIP S5 2019) is not available due to COVID-19 restrictions in field work. Were there no density data taken in the field? Usually this is the first thing that is done as it is easy and requires no high-tech equipment.

**Author response**: We do not know the details of why density of the S5 core was not measured immediately when logging the core as is standard practice. We have however triple checked that this is indeed the case with the AWI people involved in the drilling. So, we just have to accept that this data not exist. We do, however, have the seismic density data that passes right next to S5 core. So while S5 density data would be nice to have, then data would provide little extra information for validation.

As we have explained above we have a model for a correction factor, and so a validation must focus on whether it can model the density difference between sites, rather than a good density profile at a given site. The seismic density profile (Fig. 3) is near ideal for this purpose. Our model reproduces the change in BCO depth over the shear margin near perfectly, with no free model parameters. How is this is not a true way to test the model validity? We consider this to be a very strong validation of both the physical mechanism and the model sensitivity.

**Review**:
To stress that our model has to be validated in terms of changes, we have included the following paragraph in section 3:

**L250**: "Our new model for strain softening (Eq. 19) takes the form of a scale factor to classical, climate-forced models. It allows us to calculate how much faster firn densifies when the firn pack is exposed to e.g. horizontal strain rates. The quality of the fit to the overall density profile at a single site is therefore a product of both the quality of the chosen classical density model and the quality of the applied scale factor. The scale factor is widely applicable to many different firn densification models. In this paper, we want to isolate and focus on the impact of the scale factor rather than the combined effect. Equation 7 models the change in densification rate. Thus, the validation of the model must also focus on whether it is able to reproduce density changes between different strain environments. We will therefore test the predictions of the strain enhancement model on data collected at NEGIS, where large variations in horizontal strain rates, and thus in $r_h$, occur within a relatively small area."

The authors suggest the model has implications for firn densification modeling across all of Greenland and Antarctica. For example, the authors suggest the impact may be as much as 30% on the Delta-age calculated in WAIS Divide (WD). Such claims are important, and should not be made without any validation. Their claims ignore the fact that conventional firn models provide a good fit to the WD empirical Delta-age, and the WD firn density data.

**Author response**: We want to emphasize that our WAIS divide results are intended as a sensitivity test, and should not be taken as a new age model for the site. Our sensitivity test highlights the potential magnitude that strain softening could have at a site like WAIS Divide. We want to stress that we do **not** say that existing gas ages are wrong by xyz years — precisely because we realize that the WD density model is relatively tightly calibrated to local observations. This is why we carefully use words like "may", and emphasize that the estimated change in BCO age is conditional on a particular change in the horizontal strain rates.

Our results show that even a modest change in strain rate can have a substantial impact on gas age. We therefore argue that strain softening is a complicating factor in paleoclimatic interpretation. Ice flow velocities, and thus strain rates will almost certainly have been different in the past — especially at a site like WD. As an example Buizert et al. (2015) finds that "WD $\delta^{15}N$ starts to decrease around 20.5 ka BP, suggesting a thinning of the firn column". This is interpreted to be a response to early Antarctic warming. While this is completely reasonable, and we do not argue with the interpretation, then our results demonstrate that such a change could also be due to changes in ice flow.

**Review**:

To make our reasoning clear in the manuscript, we write for example:

**L481**: "Accordingly, strain softening can affect the $\Delta$age in two ways: On the one hand, it is reduced when strain rates rise. Accordingly, if the flow pattern has changed in the past, strain rates might have been different. At sites where this is the case, tuning a firn densification model to the local Holocene conditions can still induce a bias in the $\Delta$age estimates inferred for the past."

**L492**: "We stress that our WAIS Divide modeling only constitutes a sensitivity test and should not be taken as an error estimate of existing firn models which are tightly constrained by the present-day firn density profile and $\delta^{15}$N data (Buizert et al., 2015). "

At the only site where density data are shown (the EGRIP site, Fig. 2b), the "no strain" model actually provides the best fit to the data. The Antarctic map (Fig 5f) suggests several locations where the impact is 10% or more. This should easily be visible in the available firn density data (for example from South pole, EDC, EDML, Dome F and WAIS Divide).

**Author response**: Our model is a correction factor to existing densification models. It is not a complete densification model in itself. To isolate and validate the correction factor we should focus on changes to the density profile when exposed to differences in horizontal strain. We disagree that EastGRIP is the only site where data is shown. We also show the entire profile in Fig. 3, which arguably is a much stronger validation as it directly shows the change in density due to the effect we model.

We agree that the un-corrected HL model fits EastGRIP better, but it has a massive misfit in the NEGIS shear margins. So the correction is clearly a necessary improvement if you want the model to fit both sites. Further, the original empirical calibration of the HL model will account for some amount of horizontal strain as it has been calibrated to sites that are affected by some small amount of strain.

By chance the effective horizontal strain rate at EastGRIP is approximately as big as the average effective horizontal strain rate of the firn core sites that were used for tuning the HL model. It is therefore expected that the HL model provides the best fit to the data. The mismatch illustrated that the HL model needs to be recalibrated with strain softening being taken into account. For this reason we introduce $\dot{\varepsilon}_{\mathrm{cor}}$ as a first-order correction for the mismatch and the corresponding implicit strain softening contribution in the HL model.

**Review**:
**391**: "As the no-strain model already matches the data, the strain softening enhancement leads to an underestimation of the firn thickness. But by this fact, the tuning bias becomes apparent and the underestimation should not be attributed to the strain softening model, but to the underlying HL model. $\overline{\dot{\varepsilon}_{\mathrm{E,h}}}$ almost matches $\dot{\varepsilon}_{\mathrm{cor}}$. Accordingly, the firn cores used for tuning the HL model have on average experienced as much stain as the firn at EGRIP and the corresponding strain softening contribution can be expected to be already captured by the HL model. As a consequence, this contribution is considered twice, when our strain softening enhancement model is applied. In order to only consider it once, the previously introduced tuning bias correction has to be added. It increases the firn thickness again by around $7\ m$ and thereby suppresses the effect of the strain softening model."

The authors should either (1) demonstrate that their strain-enhanced model indeed improves the density data fit at various low-strain sites where such data are available, or (2) remove statements about the impact of their model on low-strain sites.

**Author response**:  We do not argue that the strain enhanced models fit low strain sites better out of the box. Most existing models probably already account for some small amount of strain softening as they have been calibrated to data from real sites (usually with low-strain rates). So the existing models may work reasonably OK at an average low strain rate site, but for the wrong reasons. Even if existing models already can fit a low strain site, then that does not mean that the strain softening is not active at the site.

Correcting models for an effect that is already included will of course not improve the fit. This does not mean that the correction factor is wrong, but rather that the existing models need to be re-calibrated with the strain enhancement accounted for, so that the model is more generally applicable. Indeed, we spend considerable space arguing that existing models need to be re-calibrated to account for this known effect.

**Review**:
See previous review.

The authors suggest a threshold (epsilon_cor = 4E-4) below which strain has no impact on densification rates. Establishing the value of this threshold seems important to how the model performs. However, the authors use a very arbitrary definition of epsilon_cor, namely the average strain rate at the calibration sites of the HL model. It seems to me that average HL value gives a lower bound estimate on epsilon_cor, but that it could easily have a much higher value. If we use epsilon_cor is 1E-3 for example, the HL model would still fit its calibration data set equally well – while presumably also fitting the NEGIS data.

**Author response**:  We do not argue that strain softening is irrelevant when $\dot{\varepsilon}_{\mathrm{E,H}} < \dot{\varepsilon}_{\mathrm{cor}}$. Rather, we argue that the original classical Herron-Langway model has been calibrated to a situation with this much strain softening on average. So, that the particular model (=HL) probably works best at sites with exactly that value.

The idea of the correction factor was to subtract this mean value from the total horizontal strain rate at a specific site in order to avoid taking this small contribution into account twice. This approach however does not work at sites with lower effective horizontal strain rates as it cannot become negative. Therefore, these values were simply fixed to 0. Nevertheless, after revising this problem now, we will change the methodology of the correction factor such that it also takes into account smaller values:

The densification rate output of the HL model can be interpreted as the densification rate by temperature/accumulation rate forcing times the scaling factor for strain softening due to an effective horizontal strain rate of $\dot{\varepsilon}_{\mathrm{cor}} = 4 \times 10^{-4} \ yr^{-1}$. To determine the densification rate for the case of no strain we therefore need to divide the densification rate give by the HL model with the $4 \times 10^{-4} \mathrm{yr}^{-1}$-scaling factor before multiplying it with scaling factor for the strain rate that is present at the site that we model. In this way we can also achieve scaling factors smaller than one at sites where the effective horizontal strain rate is lower than $4 \times 10^{-4} \mathrm{yr}^{-1}$.

In this way, the corrected strain softening model matches the HL model at EastGRIP where $\dot{\varepsilon}_{\mathrm{E,H}} \approx \dot{\varepsilon}_{\mathrm{cor}}$. Outside the ice stream – where effective horizontal strain rates are even lower – the firn thickness is now increased. Thereby the model fit is increased compared to the no-strain model which is a further indicator that the correction factor and the sensitivity of the model are accurate. The adapted correction now also has a stronger effect in the high-strain shear margins and again increases the firn thickness here, as well. The discussions of the correction factor in the manuscript will therefore be adapted accordingly.

**Review**:
The updated correction factor is now presented in subsection 2.5 on the "Tuning bias correction".

**Minor remarks**

- L16: "when old snow, respectively firn" The grammar seems off here. What is "respectively" mean here?

    **Author response**: fixed.

- L29: remove "isotope"

    **Author response**: fixed.

- L29: Delta-age is determined by the ice age at the lock-in depth, not the BCO

    **Author response**: That is correct and we have rephrased the sentence to be more clear on this point. Nonetheless, the age at the lock-in depth and the BCO age are closely related. The lock-in depth is affected by the occurrence of high-density layers that are caused by seasonal variability of precipitation and surface densities. As we do not take these into account, we do not model the lock-in depth and therefore look at BCO age instead.

**Review**:

"This age difference is primarily determined by the **age of the firn at the lock-in depth, where the air in the firn pores loses contact with the atmosphere (e.g. Schwander et al., 1997; Buizert et al., 2015). The age at the lock-in depth is again closely related to the BCO age of the firn at the firn-ice transition.**"

- L40: some models further consider grain size/growth (such as Arthern) or dust loading (such as Breant)

    **Author response**: We mention impurity content as an additional possible input parameter now. Grain size and growth however are again parameterized by temperature and accumulation rate and would fall under physical descriptions of firn processes.
    As the majority of the classical models still only takes temperature and accumulation rate into account, we keep denoting them as climate-forced.

    **Review**:

    "Despite the different approaches, **the majority of** the existing models have in common that they merely consider temperature and accumulation rate as variable input parameters (Lundin et al., 2017). **Only a few models take additional input parameters such as the impurity content into account (Bréant et al., 2017).**"

- L78: Replace "respectively" with another word.

    **Author response**: fixed.

- L108: Firn exhibits very strong horizontal density layering. So isotopic is not really true.

    **Author response**: We agree and only meant the crystallographic isotropy. Currently the sentence is removed after changes for R1.

    **Review**:
    Sentence has been removed for reviews for R1.

- L110: where did the exponent $n$ go?

    **Author response**: The exponent $n$ went into the nonlinear strain dependence of the viscosity. See Eq. 8.

- L221: The firn data in Fig. 2b suggest a lower value of $< 300 \text{kg/m}^3$

**Author response**:   For the model runs in the NEGIS region the surface density is changed to 295m. This is approximately the density of the top 0.1m observed by Schaller et al. (2016) at EastGRIP.

Fig. 2b still shows lower density values observed in the firn core. Near the surface – where the firn is still soft – material of the core is easily lost before weighing the core section and hence a bias of the density towards lower values is plausible. Therefore, we follow the dedicated surface density studies by Schaller et al.

**Review**:
**L345**: "At NEGIS the surface density is set to $295\,kg\,m^{-3}$, which is the density measured in the top $10\,cm$ in this region by Schaller et al. (2016)."

- L260: again, "respectively" is used in a way I don't understand

  **Author response**: fixed.

- L264: data is plural ("are not available")

  **Author response**: fixed.

- L267: This is an interesting observation. At even higher strain rates, do you obtain a kink in the opposite direction? This is a good target for model validation.

  **Author response**: Yes, for very high strain rates (at approx. $\dot{\varepsilon}_{E,H} > 10^{-2}$ yr$^{-1}$) our model predicts a kink in the opposite direction. The kink is however rather week and might require firn density measurements with high accuracy to be unambiguously observable. A more pronounced kink potentially requires unrealistically high strain rates as the sensitivity of firn densification to strain softening decreases at high strain rates. To our knowledge at the moment no firn density data from a region with sufficiently high strain rates exist.

- L298: I am confused. What data and what model? I assume this is fig. 3b?

  **Author response**: Correct. We will add that figure reference.

  **Review**:

  **L441** "The main difference between data and model **in Fig. 3b** lies in an apparent shift of the modeled firn density profile of 2 $km$ towards south-east."

- L299: remove comma after "No"

  **Author response**: fixed.

- L300: is the movement of the ice stream a reasonable explanation here? The firn is only 200 years old, so that would imply a lateral movement of 2km / 200 yr = 10 m per year (ballpark estimate). That seems like a lot to me – please comment.

**Author response**: We know of two other EastGRIP manuscripts currently in the works which demonstrate that the shear margin is not stable now, and has not been stable in the past (using completely different data). So, Yes. With this in mind it seems very reasonable to us. Especially considering there are no other candidates for a physical mechanism that could explain the shallow firn thickness at x=7km. We cannot cite these new studies yet, and so are careful to only offer it as a potential explanation. Our only other candidate explanation would be that there is a misalignment of the gps-positions and the seismic data. We have however, double checked this with Riverman and she assures us that it is correct.

- L322: What kind of firn air processes are you talking about? Firn air diffusion?

    **Author response**: Yes. It could be diffusion and gravitational fractionation. But we are sure that there are other firn air processes that we have not thought of that could benefit from having a natural laboratory like this: Two neighboring sites with essentially the same surface climate, but with different densification profiles. We do not want to be too specific in this sentence as this is beyond our expertise. We just want to highlight the potential because we hope that somebody will think of ways to exploit this in future work.

- L332: Is that indeed the horizontal strain rate at WAIS Divide, or is this just an example?

    **Author response**: We only have the strain rates derived from the ice sheet wide velocity fields. It is a value representative of the immediate vicinity of WAIS. However, there is some of scatter when looking at the pixels around WAIS. Further, the WAIS core is of course influenced by upstream past strain rates rather than the strain rate right at WAIS. For that reason we do not claim that our value is the WAIS strain rate, and instead frame the entire WAIS analysis as a sensitivity test.

    **Review**:
    "To gauge the potential impact of strain softening at this site, we test the sensitivity of the firn age to an **exemplary** effective strain rate of $1 \times 10^{-3} \, yr^{-1}$ using WAIS Divide climate conditions."

- L333: this is much larger than the WD Delta-age, which is 205 years at present as determined from firn air sampling (Battle et al., 2011). The WD firn density model by Buizert et al. (2015) fits this empirical constraint well, suggesting strain can safely be ignored at WD. To get a meaningful Delta-age the authors should use the ice age at the lock-in depth, and correct for the gas age at that depth.

    **Author response**: Those data only shows you that you can ignore strain softening at present. Partially because existing models implicitly account for a small amount of strain softening. However, the flow regime at WD could easily have been different in the past. Our point is that this has to be considered when modeling the past. This is what our sensitivity test shows. A small change in strain rate can have a substantial effect on BCO age (and by extrapolation also on gas age). Thus

you cannot simply conclude that the effect can safely be ignored at WD. That would be exactly like assuming that you can safely ignore changes in temperature, accumulation, or impurity content just because the model fits well at present.

In this paper we are not modeling the gas age, and we want to keep the manuscript focused on the correction itself. So we will not use lock-in depth. Our goal is not to make a new model of the gas age at WD. We are only making a sensitivity test. For this purpose it is entirely sufficient to look at the **change** in BCO. Our assumption is that if the BCO age changes by some percentage, then that will be reflected in a corresponding change in the gas age.

We will revise this section to stress the fact that this is a sensitivity test and not a correction to the WD gas ages. We will also highlight the difference between BCO age and gas age.

**Review**:
**L468**: "The gas enclosed in bubbles at the **lock-in depth** is younger than the surrounding ice."

**L478**: "For Holocene climate we find that an effective strain rate of $1 \times 10^{-3}\, yr^{-1}$ reduces the BCO age, **which we use as an approximation of $\Delta$age,** by 23 %..."

- L334: But his changes the Delta-age in the wrong direction. To obtain a smaller bipolar phasing (122 vs. the original 219) would require making WD Delta-age LARGER. So the strain correction you suggest works in the exact opposite direction of the observed correction.

  **Author response**: This is an interesting point, and we plan to add a sentences to highlight this. However, the strain softening correction can work both ways. It depends on whether you go from low-to-high, or from high-to-low strain rates. Could past strain rates be smaller than present. The point of this entire section is to make a sensitivity test.

  We will revise the section to make it absolutely clear that this is a sensitivity test, and that the results are just intended to gauge whether the effect should be considered or not. We are not making a new model of WD.

  **Review**:
  **L489**: "The shift of $\Delta$age can thereby be positive or negative. A climate forced model which has been calibrated to the locally observed density profile will implicitly account for the local present-day horizontal strain rate. Depending on whether past horizontal strain rates were greater or smaller than at present, the strain softening correction to such a model can also work in either direction."

- There are firn density data available for WAIS Divide. I would recommend you try to actually fit the firn density data (and the observed empirical Delta-age) before claiming that the established Delta-age of an ice core is incorrect by 33%.

  **Author response**: First we want to stress that we are NOT claiming that the delta age is wrong by 33%. We have apparently not been sufficiently clear that this

is only a sensitivity test. Our intent is to demonstrate that the effect can potentially be large, and must be considered. We will revise this section very carefully to ensure that this is 100% clear.

It would make sense to fit WD data if we were trying to make a new gas age model for WD. But we are not. As we argue elsewhere: Fitting the WD firn profile would not be a test of the strain softening correction, but just a test of whether Herron-Langway fits. For that reason it would strongly deviate from the focus to add the WD data to this paper.

- Line 337: again, this is in the wrong direction. To reduce the Greenland-WD phasing, one would need to INCREASE the WD Delta-age

    **Author response**: It can be either direction depending on whether past strain rates were greater or smaller than present-day strain rates. Further, it is interesting and important regardless of the direction.

    But we agree that this is a really important observation: "To reduce the Greenland-WD phasing, one would need to INCREASE the WD Delta-age" - We will revise the manuscript to explicitly highlight this.

- Line 341: Note that Buizert et al. (2021) also rely on borehole thermometry, and that the past temperature estimates from both methods agree well.

    **Author response**: We are not saying that the Buizert2021 approach does not give reasonable estimates. We are only saying that strain softening is a complicating factor (like impurity loading). The really nice thing is that if the two methods agree then that should place limits on how large the strain rates can possibly have been in the past.

    **Review**:
    **L500**: "While the results by Buizert et al. (2021) agree with the temperature inferred from borehole thermometry, our observations highlight that in regions with a strong dynamical history strain softening needs to be considered. The opposition of these observations can however also be used to infer an upper limit for the past strain rates, if the firn densification-derived temperatures are backed up by an independent method."

- Line 341: What do you expect strain rates during the LGM to be like? I would expect them to be lower, as the acc rates, surface slopes and velocities are all expected to be lower in the interior. In that case, wouldn't this make the firn column thicker during the LGM? To fit the data constraints with a more viscous firn column, one would have to make the LGM temperatures even warmer than Buizert et al. (2021) do.

    **Author response**: We agree that this is a reasonable speculation. Lower accumulation must tend to give smaller velocities just from a naive flux balance consideration. If that is the case then strain softening would be reduced in the LGM compared to present. So if that speculation holds then the reduction in strain

softening would increase the WD Delta-age and thus reduce the Greenland-WD phasing.

If we look further back in time, then we know that the bottom of the WD core is quite young, and there are some studies that suggest that WAIS might have been much smaller during the LIG. Those observations suggests that the flow patterns at WD could have been very different in the oldest part of the core. That could potentially have a large effect on the strain softening.

**Review**:
**L498**: "The large-scale ice flow could have changed over time, which complicates the modeling of past densification rates. Indeed, accumulation changes must be accompanied by changes in ice flow speeds, and thus strain rates, in order to maintain flux balance."

- L349: Can you please clarify your approach? Are you making some kind of look-up table to then interpolate to get the values in the GIS an AIS? Why not just use the gridded spacial forcing and strain rates to force the model?

  **Author response**: The two methods are equivalent. We do it this way as it is much more computationally efficient.

  **Review**:
  **L505**: "For this purpose, a range of steady-state firn density profiles are modeled with the HL model and the strain softening extension, but without the tuning bias correction being applied, to create a data grid that can be used to obtain the approximate change of the BCO depth and BCO age by strain softening at every point on the ice sheet in a computationally efficient manner by interpolation."

- L404-405: This important conclusion is based only on the model, but not validated with any data. I think validation is necessary before making claims about the validity of the method outside of high-strain environments.

  **Author response**: The fact that the sensitivity at high strain rate sites is accurate constrains the sensitivity at low strain rates. We know the effect is zero at zero strain rate and we have validated that the sensitivity is good at high strain rate as we are able to reproduce the peaks in Fig. 3b. The correction curve between these high and low strain rate sites has to be monotonically increasing (see Fig. 2). This therefore sets a lower boundary for the strength of strain softening at sites with lower strain rates. Moreover, it is physically reasonable that the sensitivity decreases with increasing strain. The driving force of densification is the load of the overlying ice. Strain softening only speeds up the process. Because the load needs to build up first, the maximum reduction of firn thickness by strain softening must be limited. Physical reasoning now tells that the corresponding value must be approached asymptotically and that the sensitivity must be decreasing.
  We can also verify whether the low strain rate sensitivity is reasonable by examining if we can reproduce the BCO contour gradient between $x = 17$km (where $\dot{\varepsilon}_{E,H} \approx 0$)

and $x = 23$km (where $\dot{\varepsilon}_{E,H} \approx 1\mathrm{kyr}^{-1}$) while $\dot{b}$ is relatively constant (Fig. 3b). In our opinion this is a good fit and demonstrates that the sensitivity of the model at low strain rates is adequate.

**Review**:

We highlight the accuracy of the model now based on the new tuning bias correction, after noting the close fit in the low-strain areas in the density profile in Fig. 3b:

**434**: "Outside the ice stream, where effective horizontal strain rate forcing is even lower than $\dot{\varepsilon}_{\mathrm{cor}}$, the firn thickness is increased. Thereby, the fit of the tuning bias corrected strain softening model is not only better in the high-strain shear margins, but also in the low-strain sections of the profile. This indicates that the tuning bias correction and the sensitivity of the model are accurate."

- L417: what do you mean be "synchronizing with gas isotopes"? Do you mean d18O-O2? This is unclear to me. synchronization is often done with CH4, not with isotopes.

  **Author response**: Agreed. This sentence will be fixed.

- Fig 2b: Your value of the surface density is too high.

  **Author response**: The surface density for this plot is changed to 295kg/m$^3$. See reply to L221.

- Fig. 5d: why are there white patches in the Antarctic firn thickness? Did the climatic conditions go outside your look-up table? Please fix.

  **Author response**: Fixed for the EAIS. The white patches in the polar hole are caused by a lack of ice velocity data at these locations.
* * *

[revised manuscript text omitted]

---

## Referee Report (RR1)

Second review for **tc-2021-240** *"Modeling enhanced firn densification due to strain softening"* by Oraschewski and Grinsted

I would first like to thank the authors for their careful reply to all the comments made during the first round of review. I think the paper has significantly been improved but there are still a number of points regarding the model development that are a bit obscures. They are listed below by order of appearance in the manuscript, as well as some minor remarks.

- lines 54, 261: I don't think theer is the need of a capital letter after a colon.

- line 98: you should start to give the exact definition $\dfrac{1}{\rho}\dfrac{\partial \rho}{\partial t} = \mathrm{tr}(\dot{\boldsymbol{\epsilon}})$

- Equation (5): the way Eq. (5) is obtained looks like a magic trick! Why not just saying that you are making the assumption that $\dot{\epsilon}_h = \dot{\epsilon}_{xx} + \dot{\epsilon}_{yy} = 0$ instead of adding it to one term to subtract it right after? This part is a bit confusing and it is difficult to get a correct meaning of all these different terms introduced for $\dot{\epsilon}_{zz}$.

- line 118: I am not sure that phase is adapted here! The constitutive relation is given on the form of a tensorial relation between Cauchy stress and strain rate, which can be decomposed in two sets of relation: one tensorial between deviatoric stress and deviatoric strain-rate and one scalar relation between isotropic pressure and rate of change in volume. Phase in a classical meaning is more referring to ice and air for example for snow, which is not the meaning here I guess?

- line 125: it should be mentioned that the two factors $a$ and $b$ are function of the density. The way it written here using "factor" makes think there are constants.

- line 170 $k_1$ and $k_2$ are not written the same way as in (10) and (11).

- line 172: not clear what you mean by the two versions of Eq. (12)?

- line 189: the fact that it is difficult to estimate the strain-rate components should be emphasized. Instead of *In summary, all components of* could be *In summary, if all components of* for example.

- lines 303, 342: I think there is a confusion about temperature, which is taken constant and uniform. If I understand that seasonal variation of temperature can be neglected, changes in temperature with depth should be accounted for, which seems not the case here? In the CFM paper, it is mentioned that $T$ is the temperature of a specific parcel of firn and thus is not uniform (function of depth and eventually of horizontal coordinates also). Is there an existing temperature profile at EGRIP that could confirm that the temperature is uniform in the firn?

- line 355: why not $295$ kg m$^{-3}$ as mentioned above?

- line 387: indicated

- Figure 5: I don't understand why all the density measurements in Greenland and Antarctica (the red dots in Fig. 5) are not compared with the model results for this last application?

---

## Author Response (AR2)

**Author's point-by-point response to the second review of tc-2021-240 by reviewer 1.**

Dear Editor Florent Dominé, dear Reviewers

we are very thankful to reviewer 1 for again taking the time to thoroughly review our manuscript. The additional comments were of great value for improving the understandability of the manuscript at places, where it was still difficult to follow. We hope that with the new changes the theory behind our model became clear.

We thank Florent Dominé for his effort and support in editing this manuscript. We further thank both reviewers for their constructive comments, which clearly helped to improve the quality of the manuscript.

Best regards,
Falk and Aslak
* * *
**Minor remarks**

- lines 54, 261: I don't think theer is the need of a capital letter after a colon.

  **Author response**: fixed.

- line 98: you should start to give the exact definition $\frac{1}{\rho}\frac{\partial \rho}{\partial t} = \mathrm{tr}(\dot{\varepsilon})$

  **Author response**: Addressed together with the following comment.

- Equation (5): the way Eq. (5) is obtained looks like a magic trick! Why not just saying that you are making the assumption that $\dot{\varepsilon}_h = \dot{\varepsilon}_{xx} + \dot{\varepsilon}_{yy} = 0$ instead of adding it to one term to subtract it right after? This part is a bit confusing and it is difficult to get a correct meaning of all these different terms introduced for $\dot{\varepsilon}_{zz}$.

  **Author response**: We agree that the derivation of the general expression of the strain softening correction is difficult to follow. We therefore reformulated the section by starting with the general definition of the strain rate tensor and its relation to the densification rate (previous comment) as well as by giving explicitly the assumptions needed for disentangling horizontal divergence and strain softening (this comment). While the following changes look severe, we mainly changed the order of equations, so that the general definition of the relation between strain rate and densification rate is understandable when it is introduced and to then keep a logical structure after doing so.

[revised manuscript text omitted]

- line 118: I am not sure that phase is adapted here! The constitutive relation is given on the form of a tensorial relation between Cauchy stress and strain rate, which can be decomposed in two sets of relation: one tensorial between deviatoric stress and deviatoric strain-rate and one scalar relation between isotropic pressure and rate of change in volume. Phase in a classical meaning is more referring to ice and air for example for snow, which is not the meaning here I guess?

  **Author response**: Here, we follow the terminology of Duva and Crow (1994), who write:

  > "A key feature of our modelling is that the reinforced porous material is regarded as a two phase material consisting of a rigid reinforcing phase and a homogeneous, porous creeping phase."

  To be consistent with the original literature, we prefer to keep the wording as it is, but add a sentence, that these phases shall not be confused with the ice and air phase that are often used for describing firn.

**Review**:

**L122**: "Firn is a compressible material. **Following Duva and Crow (1994),** its deformation is given by two coupled constitutive relations which represent a rigid **reinforcing phase** and a compressible phase**, that are not to be confused with the ice and air phases often used for describing firn. The rigid phase is defined** by the **tensorial** relation between deviatoric stress and deviatoric strain rate, as given by Glen's flow law for the deformation of incompressible ice (Glen, 1955; Nye, 1957). The **scalar** relation of the compressible phase is set between the **isotropic pressure and the volumetric strain rate**."

- line 125: it should be mentioned that the two factors $a$ and $b$ are function of the density. The way it written here using "factor" makes think there are constants.

  **Author response**: We agree and changed the term "weighting factors" to "density dependent weighting coefficients".

- line 170 $k_1$ and $k_2$ are not written the same way as in (10) and (11).

  **Author response**: fixed.

- line 172: not clear what you mean by the two versions of Eq. (12)?

  **Author response**: We explicitly write it out now to be clear.

**Review**:

**L173**: "Equations 8 to 12 are applicable independent of whether the additional strain rate components are considered. They can be formulated both in terms of the strain softening corrected model and the climate-forced model, with the only difference being whether the macroscopic effective strain rate is computed from Eq. 2 or 4. In particular, $k_1$, $k_2$ and $\tau_{\mathrm{ice},zz}$ do not differ between the two cases as they are assumed to be independent of the additional strain rate components. **Thus, Eq. 12 can analogously be formulated for the climate-forced model case as**

$$\dot{\varepsilon}_{zz,\mathrm{c}} = \frac{k_2}{2\eta_{\mathrm{ice,c}}}\tau_{\mathrm{ice},zz},\tag{13}$$

**where the effective ice viscosity in the climate-forced model $\eta_{\mathrm{ice,c}}$ will be higher than the actual effective ice viscosity $\eta_{\mathrm{ice}}$.**

**2.3 The scale factor**

By dividing **Eqs. 12 and 13**, the strain rate ratio in Eq. 7, which is the scale factor, can be expressed in terms of a solid ice viscosity ratio:

$$\frac{\dot{\varepsilon}_{zz}}{\dot{\varepsilon}_{zz,\mathrm{c}}} = \frac{\eta_{\mathrm{ice,c}}}{\eta_{\mathrm{ice}}}.\tag{14}$$

"

- line 189: the fact that it is difficult to estimate the strain-rate components should be emphasized. Instead of *In summary, all components of* could be *In summary, if all components of* for example.

    **Author response**: We agree and change the paragraph accordingly. Also we add a sentence saying that even knowing some of the components (e.g. the horizontal strain rates) and correcting for it improves the estimate of the densification rate.

    **Review**:
    **L198**: "In summary, the variable $r_{\mathrm{v}}$ corresponds exactly to the scale factor that is sought. **If all strain rate components in $r_{\mathrm{h}}$ are known, only Eq. 17 is left** to be solved for $r_{\mathrm{v}}$ to obtain the scale factor for correcting the densification rate of a climate-forced model for the total effect of strain softening. **But even if merely some of the external strain rate components in Eq. 19, e.g. the horizontal strain rates, are known, this approach can be used to correct for their contribution to strain softening enhancement of the densification rate.**"

- lines 303, 342: I think there is a confusion about temperature, which is taken constant and uniform. If I understand that seasonal variation of temperature can be neglected, changes in temperature with depth should be accounted for, which seems not the case here? In the CFM paper, it is mentioned that $T$ is the temperature of a specific parcel of firn and thus is not uniform (function of depth and eventually of horizontal coordinates also). Is there an existing temperature profile at EGRIP that could confirm that the temperature is uniform in the firn?

    **Author response**: The CFM model does allow for variable temperature forcing, and so it is simple to apply a variable temperature in the model. However, in practice firn temperatures below 10 m change very little (See e.g. Dahl-Jensen et al. (1998) or Orsi et al. (2017)). This is because vertical velocities are relatively large near the surface and downward advection of cold surface temperatures overwhelms any heating from below. For the same reason, Herron and Langway had so much success with a firn model that uses the 10 m temperature as the only temperature input. Nonetheless, a recent warming trend can be observed in the firn temperatures in Greenland, but with it being only on the order of 1°C its affect on firn densification is minor and can be neglected. This is especially the case regarding the purpose of our work in which we do not want to produce exact estimates of certain firn properties, but to study the general effect of strain softening on firn densification.

    **Review**:
    **L313**: "Temperature evolution **is neglected** in our **model** experiments, as **we aim to assess the general impact of strain softening on firn densification and thereby study processes occurring** in the second firn stage, **where temperature is approximately stable. At this depth** seasonal temperature variations are dampened by heat conduction **and only a recent warming trend**

**remains, which for North Greenland lies on the order of** $1°C$ **(Orsi et al., 2017) and, hence, has a minor impact on firn densification.**"

**L352**: "We force the model with a constant temperature of $-29.9°C$. This is the seasonality-corrected mean of the 10 m temperature recorded between June 2019 and January 2021 at the PROMICE weather station at EGRIP (Fausto and van As, 2019; Fausto et al., 2021). Using a constant temperature input is justified because we are mainly interested in firn processes occurring below a depth of 10 m, where seasonal variability of temperature is smoothed out by heat conduction **and the impact of the general warming trend in Greenland is minor**. Further, we do not expect a significant spatial variability of temperature over this relatively small study region."

- line 355: why not $295\,\mathrm{kg\,m}^{-3}$ as mentioned above?

  **Author response**: We changed this to accommodate an earlier review. The effect of such a small change is minimal on the deeper density profile, as the densification is very rapid near the surface, so we have decided to keep it as it is, but now start the sentence with a clarification, that we are specifically referring to the ice sheet wide studies.

  **Review**:
  **L370**: "**In the ice sheet wide studies,** new surface layers are formed with a density of $315\,\mathrm{kg\,m}^{-3}$, following Fausto et al. (2018)."

- line 387: indicated

  **Author response**: fixed.

- Figure 5: I don't understand why all the density measurements in Greenland and Antarctica (the red dots in Fig. 5) are not compared with the model results for this last application?

  **Author response**: This may sound like a small task, but it really require a full (separate) calibration-validation study before this would be useful. We would estimate this would add many pages of text, new data sets, and multiple figures. A lot of work that we further feel would ultimately strongly detract from the focus of the study. So, we leave this for future work - In our opinion - this is simply beyond the scope of the present study. As we have said before, there is no advantage in comparing a wrong model to the data. The dots are there to illustrate that strain softening has some effect at the sites that were used to tune the model. We make that more clear in the figure caption now.

  **Review**:
  "**Figure 5.** Study on the contribution of strain softening to firn densification in terms of the firn thickness over the dry zone of the Greenland ice sheet (GrIS, a-c) and the Antarctic ice sheet (AIS, d-f). (a & d) Modeled total firn thickness when strain softening is considered without the tuning bias correction. (b & e)

Absolute firn thinning contribution caused by strain softening. (c & f) Relative firn thinning due to strain softening, which also illustrates by how much the densification process is accelerated by strain softening. Red dots indicate the drill locations of the firn cores that were used for tuning the empirical Herron-Langway model **and illustrate that some of these firn cores were considerably affected by strain softening**. Outside the dry zone of GrIS the surface velocity is shown with brighter colors indicating faster flow."
* * *